



Geoscientific
Model Development

# The road weather model RoadSurf (v6.60b) driven by the regional climate model HCLIM38: evaluation over Finland

**Erika Toivonen**[1], **Marjo Hippi**[1], **Hannele Korhonen**[1], **Ari Laaksonen**[1,2], **Markku Kangas**[1], **and Joni-Pekka Pietikäinen**[1,a]

[1]Finnish Meteorological Institute, Helsinki, Finland
[2]Department of Applied Physics, University of Eastern Finland, Kuopio, Finland
[a]now at: Climate Service Center Germany (GERICS), Hamburg, Germany

**Correspondence:** Erika Toivonen (erika.toivonen@fmi.fi) and Joni-Pekka Pietikäinen (joni-pekka.pietikaeinen@hzg.de)

**Abstract.** In this paper, we evaluate the skill of the road weather model RoadSurf to reproduce present-day road weather conditions in Finland. RoadSurf was driven by meteorological input data from cycle 38 of the high-resolution regional climate model (RCM) HARMONIE-Climate (HCLIM38) with ALARO physics (HCLIM38-ALARO) and ERA-Interim forcing in the lateral boundaries. Simulated road surface temperatures and road surface conditions were compared to observations between 2002 and 2014 at 25 road weather stations located in different parts of Finland. The main characteristics of road weather conditions were accurately captured by RoadSurf in the study area. For example, the model simulated road surface temperatures with a mean monthly bias of $-0.3\,°C$ and mean absolute error of $0.9\,°C$. The RoadSurf's output bias most probably stemmed from the absence of road maintenance operations in the model, such as snow plowing and salting, and the biases in the input meteorological data. The biases in the input data were most evident in northern parts of Finland, where the regional climate model HCLIM38-ALARO overestimated precipitation and had a warm bias in near-surface air temperatures during the winter season. Moreover, the variability in the biases of air temperature was found to explain on average 57 % of the variability in the biases of road surface temperature. On the other hand, the absence of road maintenance operations in the model might have affected RoadSurf's ability to simulate road surface conditions: the model tended to overestimate icy and snowy road surfaces and underestimate the occurrence of water on the road. However, the overall good performance of RoadSurf implies that this approach can be used to study the impacts of climate change on road weather conditions in Finland by forcing RoadSurf with future climate projections from RCMs, such as HCLIM.

## 1 Introduction

The road traffic sector is one field benefiting from improved regional weather and climate information, especially at northern high latitudes. These regions do not only experience frequent wintertime snow and ice conditions but also rapidly changing road weather due to, for instance, the onset of snowfall (Juga et al., 2012) or during temperature variations around the freezing point (Kangas et al., 2015). Systematic consideration of upcoming weather events helps the general public in their everyday commute and, furthermore, road maintenance authorities to attend the roads in a cost-effective manner (Nurmi et al., 2013). In Finland, the Finnish Meteorological Institute (FMI) has a duty to issue warnings of hazardous traffic conditions to the general public. To support this, the institute has developed a road weather model, RoadSurf, which has been in operational use since 2000 (Kangas et al., 2015).

Road weather conditions are expected to be affected by ongoing anthropogenic climate change (e.g. Jaroszweski et al., 2014) throughout the inhabited northern high latitudes. This region is strongly impacted by the Arctic amplification of climate warming (Screen, 2014), which can clearly be seen, for instance, in the Finnish temperature records of the past 170 years (Mikkonen at al., 2015). The expected

warmer and wetter future climate implies new challenges for road maintenance and traffic safety, especially in the southern parts of Finland: precipitation events are likely to shift towards less snowfall and more frequent rain and sleet episodes (Räisänen, 2016). This kind of change in climate will decrease snowy road conditions but at the same time increase the occurrence of wet road surfaces, which could lead to more frequently observed slippery and icy road conditions during the coldest times of a day, such as nighttime (Andersson and Chapman, 2011a). Moreover, the events of temperature change around the freezing point might become more frequent in the northern parts of Finland (Makkonen et al., 2014), leading to an increased occurrence of black ice conditions and making the roads more vulnerable to erosion. Therefore, policymakers and other stakeholders should have access to credible regional climate projections that can provide a solid basis for informed impact assessments and adaptation measures in the road weather sector. A central tool for producing such projections is high-resolution regional climate models (RCMs).

Although the impacts of climate change on road weather, safety, and design have been assessed in many studies (e.g. see Koetse and Rietveld, 2009), most of these studies have only considered relative changes in air temperature and precipitation and related these to the possible impacts on the roads (e.g. Andersson and Chapman, 2011a, b; Hambly et al., 2013; Hori et al., 2018; Makkonen et al., 2014). It would be beneficial to study the climate change impacts on, for instance, road surface temperatures ($T_{road}$) or road surface conditions using an approach in which these impacts can be accessed more directly. Furthermore, as slippery road conditions, such as snowy or icy roads, are the major cause for the wintertime and weather-related road accidents in Fennoscandia (Andersson and Chapman, 2011b; Malin et al., 2019; Salli et al., 2008), it is essential to estimate how frequently these conditions will occur in the future.

The main goal of this paper is to evaluate the skill of RoadSurf to reproduce present-day road weather conditions in Finland when driven by a state-of-the-art high-resolution RCM, cycle 38 of the HIRLAM-ALADIN Regional Mesoscale Operational Numerical Weather Prediction (NWP) In Europe (HARMONIE) Climate (HCLIM) (Lindstedt et al., 2015). HCLIM is forced by the ERA-Interim reanalysis product (Dee et al., 2011) in the lateral boundaries since it is a standard procedure to carry out evaluation experiments using the (close to) perfect boundary settings in RCMs (e.g. Kotlarski et al., 2014). This is the first time that such a modeling chain is evaluated, and therefore this evaluation is needed in order to build and study future scenarios of road weather in this area with higher confidence. Although high-resolution climate projections for Europe are currently available through the EURO-CORDEX international climate downscaling initiative that provides RCM data at 50 km (EUR-44) and 12.5 km (EUR-11) resolutions (Jacob et al., 2014), the EURO-CORDEX dataset does not publicly

include reanalysis-driven RCM simulations at very high temporal resolutions, such as 1-hourly. Therefore, meteorological input data for RoadSurf are taken from HCLIM, which is run for the years 2002–2014 with ALARO physics (Gerard, 2007; Gerard et al., 2009; Piriou et al., 2007) at 12.5 km resolution. These HCLIM simulations are evaluated against standard meteorological datasets over Finland: E-OBS v19.0e (Cornes et al., 2018) and the ERA5 reanalysis product (C3S, 2017).

In the previous studies, mainly NWP model outputs have been used to force RoadSurf. The simulated road weather parameters, such as $T_{road}$, have been verified against observations over Finland (Karsisto et al., 2016) and the Netherlands (Karsisto et al., 2017). In addition, Kangas et al. (2015) have studied RoadSurf's ability to simulate the amount of water, snow, frost, and ice on the road (called storage terms in RoadSurf) as well as road surface conditions and friction values, although only for two road weather stations in Finland. These studies have considered relatively short verification periods varying from 1 week to some months. In this paper, we concentrate on 13-year-long simulations of HCLIM and HCLIM-driven RoadSurf. First, the performance of HCLIM is evaluated by comparing the model results with E-OBS v19.0e dataset of near-surface air temperature and precipitation and with ERA5 reanalysis for downwelling shortwave and longwave radiation, total cloud fraction (clt), relative humidity, and wind speed. All of these parameters, excluding clt, are used as inputs for RoadSurf. This comparison is followed by an evaluation of HCLIM-driven RoadSurf against observations at 25 road weather stations located in Finland. The focus is on $T_{road}$ but also the simulated road surface conditions and storage terms are compared to the observations. In addition, this study investigates the role of the biases in the HCLIM data on the biases in road surface temperature produced by HCLIM-driven RoadSurf.

## 2 Models and data

### 2.1 Models

#### 2.1.1 HARMONIE-Climate (HCLIM)

HARMONIE is a seamless NWP model framework developed in collaboration with several European national meteorological services (Bengtsson et al., 2017). The nonhydrostatic and spectral dynamical cores in HARMONIE are provided by ALADIN–NH (Bénard et al., 2010), which solves the fully compressible Euler equations using a two-time level, semi-implicit, semi-Lagrangian discretization on an Arakawa A grid. This study applied a model setup using the cy38h1 climate model version of HARMONIE with ALARO physics (HCLIM38-ALARO hereafter), as mentioned before; a hydrostatic version of the dynamical core; and a time step of 300 s. The HCLIM38-ALARO version used in this

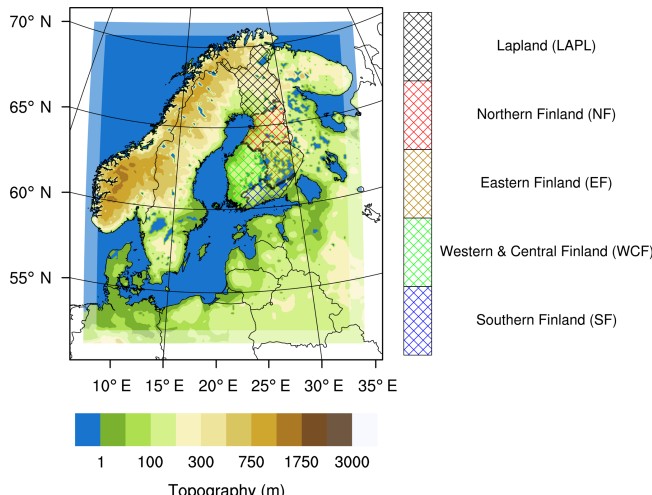

**Figure 1.** The HCLIM38-ALARO model domain and topography at 12.5 km × 12.5 km grid resolution. Colored overlays depict the regions that are evaluated in more detail. The transparent areas depict the model's 8-point wide relaxation zone.

study includes a lake model, FLake (Mironov, 2008; Mironov et al., 2010), and a surface parameterization framework, surface externalisée (SURFEX) (Masson et al., 2013). A more thorough description of HCLIM can be found in Lindstedt et al. (2015).

For this study, HCLIM38-ALARO was run from January 2002 to December 2014 (years 2000 and 2001 as a spinup) over the Fennoscandian domain (151 × 181 grid boxes) with 12.5 km × 12.5 km horizontal grid resolution and 65 vertical layers. Figure 1 depicts the HCLIM38-ALARO simulated domain along with the model's 8-point wide relaxation zone as well as the regions of Finland that are analyzed in more detail in this study. The sea surface (sea-surface temperature and sea-ice concentration) and lateral boundary conditions of HCLIM38-ALARO were taken from ERA-Interim reanalysis (Dee et al., 2011) every 6 h. In this study, the HCLIM38-ALARO output parameters were produced every full hour and were used to force RoadSurf offline.

### 2.1.2 RoadSurf

The road weather model RoadSurf used in this study is a 1-D model based on solving the energy balance at the ground surface. This study employed the RoadSurf version 6.60b, which is the operational version of the FMI's research department with slight I/O changes made for this study. The model takes into account the conditions at the road surface and beneath it, and calculates the vertical heat transfer into the ground as well as at the interface of ground and atmosphere. Hydrological processes, such as accumulation of rain and snow, run-off from the surface, sublimation, freezing, melting, and evaporation, are parameterized. The model esti-

mates road surface friction using a numerical statistical equation (Juga et al., 2013). RoadSurf assumes a flat horizontal surface which does not have any shading elements, such as trees. However, topography in general is taken into account implicitly through the input data. Thermodynamic properties of the road surface and ground are assumed to be similar for all simulated points, and the first two layers of the surface are always described as asphalt. In addition, the effect of traffic on the road surface is included: the model assumes that traffic packs some part of the snow into ice, whereas the remaining part is assumed to be blown away from the road. However, the model does not take into account wintertime road maintenance operations, such as salting and snow plowing, because RoadSurf is also used to plan and optimize these maintenance actions. The absence of road maintenance in the model implies that there will be unavoidable discrepancies when comparing the modeled and observed road weather conditions.

As inputs, RoadSurf needs near-surface air temperature ($T_{air}$), near-surface relative humidity (RH), 10 m wind speed (WS), precipitation (Pr); and downwelling shortwave ($SW_d$) and longwave ($LW_d$) radiation. In the operational use, the model employs observations from road weather stations, meteorological SYNOP (surface synoptic observations) weather stations, and radar precipitation networks to initialize road conditions while the road weather is predicted for the upcoming days utilizing forecasts produced by NWP models. In this study, we did not include any forecasted periods implying that no in situ observations were used to initialize and force RoadSurf. Instead, RoadSurf was modified so that it utilizes the RCM data, in this case the output of reanalysis-driven HCLIM38-ALARO. In addition to the abovementioned inputs needed by RoadSurf, we utilized the bottom layer ground temperature (at the depth of 4.28 m) produced by HCLIM38-ALARO. Using the simulated ground temperature instead of the climatological one was motivated by the fact that although in the original RoadSurf version this temperature is assumed to vary sinusoidally, it is estimated by an equation in which some of the parameter values are based on measurements retrieved from only one FMI observatory located in Southern Finland. RoadSurf's main outputs are $T_{road}$ and a traffic index describing driving conditions, but the model produces also surface friction; prevailing road conditions; and the sizes of water, snow, and ice storages on the road. RoadSurf divides the road surfaces into eight classes: dry, damp, wet, wet snow, frosty, partly icy, icy, and "dry snow". This classification is mainly based on the storage terms and $T_{road}$. The model physics of RoadSurf is described in more detail in Kangas et al. (2015).

## 2.2 Evaluation data

### 2.2.1 E-OBS dataset of gridded daily precipitation and near-surface air temperature

The HCLIM38-ALARO simulated daily precipitation and near-surface air temperatures were compared with the E-OBS dataset, version 19.0e (Cornes et al., 2018), which consists of daily precipitation and 2 m air temperature (daily minimum, mean, and maximum) data retrieved from stations located in Europe. The data are available as a regular grid which covers the pan-European domain with a resolution of 0.11° (approximately 12 km). This E-OBS version, 19.0e, consists of a 100-member ensemble of realizations for each daily field. We utilized ensemble means that can be taken as grid box averages (Cornes et al., 2018) and that are comparable to the best guess grid in the earlier versions of E-OBS (Haylock et al., 2008).

In general, gridded datasets, such as E-OBS, include uncertainties due to the use of point measurements (e.g. rain gauges) and interpolation procedures. For example, the undercatch of precipitation can lead to high biases especially in winter at high latitudes as well as in the areas of rough topography (e.g. Prein and Gobiet, 2017). These undercatch errors are typically between 3 % and 20 % for rainfall and up to 40 % (for shielded) or even up to 80 % (for nonshielded) gauges for snow (Goodison et al., 1998). Moreover, the accuracy and success of the E-OBS dataset depend on the number of stations used in the gridding process (Cornes et al., 2018): the sparse station density can introduce errors into the gridded dataset (e.g. Prein and Gobiet, 2017). For Finland, the station density is sparser in the northern parts compared to the south (Fig. S1 in the Supplement). Although these observational uncertainties are not in the scope of this study, they should be kept in mind when analyzing the results.

The comparison of modeled and observed data was performed using the coarsest grid resolution. The HCLIM38-ALARO model results covering Finland were thus compared with E-OBS by remapping the E-OBS values into the grid of HCLIM38-ALARO: temperature data by using bilinear and precipitation data by using first-order conservative remapping. The areas with a lake fraction greater than or equal to 0.5 have been excluded from the analysis because E-OBS data over the lakes are based on the interpolation of the measurements over land. Moreover, the modeled 2 m air temperature values have been corrected using a lapse rate of $0.0064\,°C\,m^{-1}$ to account for the differences between the orography in the E-OBS dataset and the model. A standard Student's $t$ test at a 95 % confidence level was used to assess the significance of the differences between the modeled and observed monthly averages (in the case of temperature) or monthly sums (in the case of precipitation).

### 2.2.2 ERA5 reanalysis product

Reanalysis is a scientific method that is based on a combination of data assimilation and numerical models. The fifth generation of the ECMWF's atmospheric reanalyses of the global climate, ERA5, provides hourly atmospheric data estimates at a horizontal grid resolution of approximately 30 km (Hersbach et al., 2018). This product was created using 4D-Var data assimilation and the ECMWF's Integrated Forecast System (IFS) cycle 41r2 that was used as the operational medium-range forecasting system in 2016. The model includes 137 levels in the vertical reaching to 1 Pa. Overall, ERA5 assimilates more observations compared to the ERA-Interim reanalysis product. However, it is good to note that the ERA5 dataset is based on a model that is assimilating observations and thus the dataset is prone to similar model deficiencies as other weather and climate models.

We utilized the monthly means of daily means for clt, $SW_d$, $LW_d$, 10 m WS, and near-surface RH to evaluate the performance of HCLIM38-ALARO. Monthly means of daily-mean RH were computed employing the ERA5 product of hourly near-surface $T_{air}$ and dew point temperature ($T_{dew}$) ($RH = 100 \times e_s(T_{dew})/e_s(T_{air})$) as RH is not archived directly in the ERA5 dataset. Saturation vapor pressure ($e_s$) was calculated based on the Magnus formula and with respect to water (WMO, 2008). Modeled near-surface RH was directly available and used as such.

Similarly to the comparison with the E-OBS data, the evaluation was carried out using the coarsest grid resolution by remapping HCLIM38-ALARO model results into the ERA5 grid using bilinear interpolation. Again, a standard Student's $t$ test at a 95 % confidence level was used to assess the significance of the differences between the modeled and observed monthly averages (in the case of clt, $LW_d$, WS, and RH) or seasonal averages (in the case of $SW_d$).

### 2.2.3 Road weather stations

The results obtained by the RoadSurf-HCLIM configuration were compared with observations retrieved from 25 road weather stations located in different regions of Finland. Table 1 describes the features of these stations, such as location, surrounding characteristics, road maintenance class, and the monthly average air temperatures, during October and April from 2002 to 2014. Stations 1–8 are located in Southern Finland, stations 9–13 in Western and Central Finland, stations 14–16 in Eastern Finland, stations 17–21 in Northern Finland, and stations 22–25 in Lapland (Fig. 2). The model grid cell closest to each of these stations was selected for evaluation. However, it needs to be noted that the model output represents an areal average over the whole model grid cell, whereas the road weather observations are point measurements.

The road weather stations are equipped with the Vaisala ROSA road weather package and Vaisala DRS511 sensors

**Table 1.** Descriptions of the road weather stations with the mean observed air temperatures (°C) for the months between October and April in 2002–2014. The stations with an optical sensor are marked with an asterisk (*). The road orientation is defined in parenthesis. As an example, SE–NW means that the orientation of the road is from southeast to northwest. The maintenance classes are described in Appendix A (class 1 means high maintenance and class 4 low maintenance).

| Region | Number | Station name | Coordinates | Surrounding characteristics and road orientation | Maintenance class | Mean $T$ (°C) October | Mean $T$ (°C) November | Mean $T$ (°C) December | Mean $T$ (°C) January | Mean $T$ (°C) February | Mean $T$ (°C) March | Mean $T$ (°C) April |
|---|---|---|---|---|---|---|---|---|---|---|---|---|
| Southern Finland | 1 | Askisto | 60.27° N, 24.77° E | Open area and a few trees (E–W) | 1 | 5.8 | 1.6 | −2.0 | −5.1 | −5.5 | −2.1 | 4.4 |
| | 2 | Lappeenranta | 61.07° N, 28.31° E | Open area and trees nearby (SW–NE) | 2 | 4.3 | 0.0 | −4.4 | −7.6 | −7.6 | −3.3 | 3.6 |
| | 3 | Sutela | 60.50° N, 26.88° E | Open area and a few trees, river nearby (E–W) | 2 | 5.6 | 1.1 | −2.4 | −5.5 | −7.0 | −2.7 | 3.8 |
| | 4* | Jakomäki | 60.25° N, 25.06° E | Open area and trees on both sides of the road (SW–NE) | 1 | 6.1 | 1.8 | −1.5 | −4.4 | −5.2 | −1.7 | 4.4 |
| | 5* | Lahti | 60.91° N, 25.61° E | Open area (SW–NE) | 1 | 4.7 | 0.8 | −3.2 | −6.5 | −6.6 | −2.6 | 4.0 |
| | 6* | Palojärvi | 60.29° N, 24.32° E | Open area and a few trees, trees on the opposite side (E–W) | 1 | 5.1 | 1.2 | −2.5 | −5.5 | −5.8 | −2.6 | 3.8 |
| | 7* | Riihimäki | 60.71° N, 24.74° E | Empty lane between the road (SE–NW) | 1 | 4.8 | 0.4 | −3.9 | −6.1 | −5.9 | −2.4 | 4.2 |
| | 8* | Utti | 60.89° N, 26.86° E | Open area, a few trees, and trees on the opposite side of the road (E–W) | 2 | 4.2 | 0.3 | −3.8 | −7.1 | −7.1 | −2.9 | 3.7 |
| Western & Central Finland | 9 | Lapua | 62.94° N, 23.04° E | Open area and trees on both sides of the road (S–N) | 2 | 3.8 | −0.3 | −4.0 | −6.9 | −6.6 | −3.0 | 3.5 |
| | 10 | Petäjävesi | 62.27° N, 25.39° E | Open area, a few trees, and trees on the opposite side of the road (E–W) | 3 | 3.5 | −0.7 | −5.0 | −8.2 | −8.2 | −4.2 | 2.7 |
| | 11 | Seppälänahde | 61.21° N, 22.45° E | Open area and trees on both sides of the road (SE–NW) | 2 | 4.9 | 0.9 | −2.8 | −5.9 | −5.7 | −2.3 | 4.0 |
| | 12* | Suinula | 61.55° N, 24.07° E | Open area and trees on both sides of the road (SW–NE) | 2 | 4.3 | 0.2 | −4.0 | −7.0 | −7.1 | −3.5 | 3.3 |
| | 13* | Vaasa | 63.14° N, 21.76° E | Open area and trees on both sides of the road (SW–NE) | 2 | 4.6 | 0.3 | −3.6 | −5.7 | −6.6 | −3.2 | 2.9 |
| Eastern Finland | 14 | Kuopio E | 62.84° N, 27.61° E | Empty lane between the road (S–N) | 1 | 3.6 | −0.7 | −5.2 | −8.9 | −8.8 | −4.1 | 2.8 |
| | 15* | Puunkolo | 62.06° N, 27.81° E | Open area, a few trees, and trees on the opposite side of the road (S–N) | 3 | 3.6 | −0.9 | −5.6 | −8.7 | −8.7 | −4.3 | 2.6 |
| | 16* | Ylämylly | 62.63° N, 29.60° E | Open area (SW–NE) | 2 | 3.8 | −1.0 | −5.7 | −9.1 | −9.2 | −4.5 | 2.3 |
| Northern Finland | 17 | Kalajoki | 64.34° N, 23.96° E | Open area and trees on both sides of the road (SW–NE) | 3 | 4.3 | −0.1 | −3.8 | −7.1 | −7.3 | −4.1 | 1.8 |
| | 18 | Korholanmäki | 64.14° N, 28.00° E | A few trees and trees on the opposite side of the road (SE–NW) | 3 | 2.3 | −2.5 | −6.6 | −9.6 | −9.4 | −4.9 | 2.0 |
| | 19 | Kuolio | 65.83° N, 28.82° E | Open area (SW–NE) | 4 | 0.9 | −4.4 | −8.6 | −12.2 | −11.6 | −7.7 | −0.5 |
| | 20 | Kärsämäki | 64.01° N, 25.76° E | Open area and trees on the opposite side of the road (S–N) | 3 | 2.7 | −1.8 | −6.1 | −9.4 | −9.0 | −4.7 | 2.1 |
| | 21 | Ouluntulli | 64.95° N, 25.53° E | Open area and a small hill nearby (SE–NW) | 1 | 3.2 | −1.4 | −5.6 | −9.1 | −8.8 | −5.1 | 2.0 |
| Lapland | 22 | Saariselkä | 68.46° N, 27.43° E | Open area (SW–NE) | 4 | −0.5 | −6.0 | −8.4 | −11.2 | −11.2 | −7.2 | −1.4 |
| | 23 | Sieppijärvi | 67.00° N, 24.05° E | Open area and trees on both sides of the road (S–N) | 4 | 0.1 | −6.7 | −9.1 | −12.8 | −11.9 | −6.9 | 0.5 |
| | 24* | Jaatila | 66.25° N, 25.34° E | Open area and trees on both sides of the road (SW–NE) | 3 | 1.6 | −4.0 | −7.4 | −11.2 | −10.6 | −6.2 | 1.0 |
| | 25* | Kyläjoki | 65.84° N, 24.26° E | Open area, at the start of an overpass (E–W) | 2 | 2.6 | −2.6 | −6.1 | −9.8 | −9.7 | −5.6 | 0.8 |

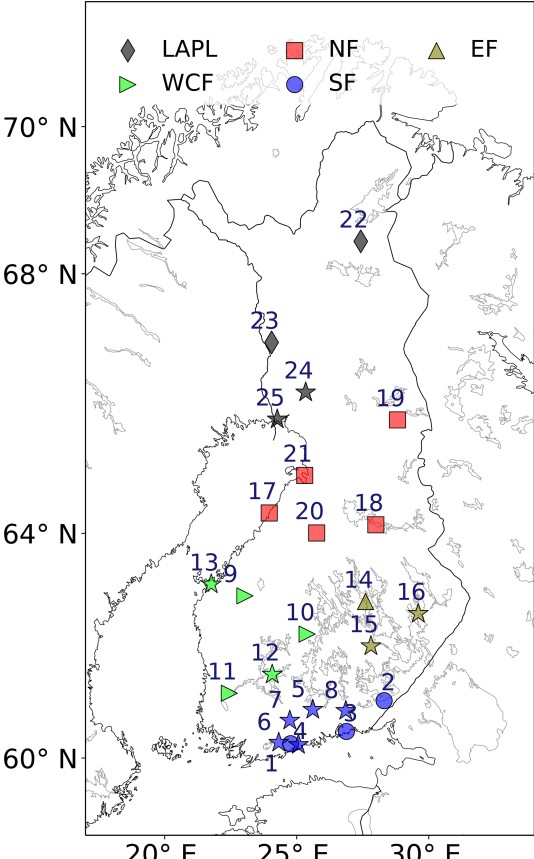

**Figure 2.** Locations of road weather stations used in this study. The numbers refer to Table 1. The stations with an additional optical sensor are marked as stars. SF stands for Southern Finland, WCF for Western and Central Finland, EF for Eastern Finland, NF for Northern Finland, and LAPL for Lapland.

(Vaisala, 2018a), which are installed in the road surface. Thirteen of the selected stations included also the Vaisala DSC111 optical sensor (Vaisala, 2018b), which provides information on, for instance, water, snow, and ice storages on the road. Two of the stations with an optical sensor had a large amount of missing data and, therefore, only 11 of them were included in this study. This study employs the road surface temperature and the information on the road surface classes provided by the ROSA stations and the storage terms provided by the stations with the additional optical sensors. Data availability was on average 79 % (range 57 %–91 %) at ROSA stations and 32 % (range 18 %–38 %) at stations with the optical sensor during the study period of 2002–2014.

The classification of observed and modeled road surface conditions differ slightly. For example, the observations included "damp and salty" as well as "wet and salty" road surface classes. These classes were combined with "damp" and "wet", respectively because RoadSurf does not include information on salting of the roads. The "wet snow" and "dry snow" classes provided by RoadSurf were also grouped to-

gether considering that observations did not have a directly comparable class for wet snow. In addition, observations do not include a "partly icy" class which is defined in the model. Therefore, these divergent definitions of road condition classes might cause some discrepancies when comparing the modeled and observed road conditions.

## 3   Results and discussion

### 3.1   Evaluation of HCLIM38-ALARO

#### 3.1.1   Mean near-surface air temperature

The HCLIM38-ALARO model accurately captured the daily and seasonal mean 2 m air temperatures ($T_{air}$) over Finland between 2002 and 2014. This is confirmed by Fig. S2 which illustrates the probability density functions (PDF) of the daily $T_{air}$ for the observations and model during different seasons over all the grid points falling over Finland. Overall, the general shapes of $T_{air}$ distributions were correctly reproduced by HCLIM38-ALARO with the largest deviations found in the winter season (December–February).

Also, the multiyear mean seasonal $T_{air}$ was well captured by HCLIM38-ALARO. Figure 3 shows the seasonal means from E-OBS as well as the mean biases in the HCLIM38-ALARO simulated mean seasonal $T_{air}$ with a reference to E-OBS. The stippled areas depict significant differences indicated by the Student's $t$ test ($p < 0.05$). The mean biases averaged over Finland were slightly positive in the autumn and winter (September–February) and negative in the spring and summer (March–August). The autumn season had the smallest domain-averaged bias of 0.004 °C and the summer season the highest domain-averaged bias of $-0.40$ °C. The biases were statistically significant mainly over the northern parts of Finland where the model had an enhanced warm bias in the winter and cold bias in the summer. These biases might partly be caused by the lower station density in the northernmost domain, which might decrease the accuracy of the E-OBS data. On the other hand, the model was in good agreement with the observations during the spring and autumn when most of the differences were not statistically significant.

It is good to note that Lindstedt et al. (2015) encountered similar warm biases in their HCLIM-ALARO simulations with cycle 36 over Sweden during the wintertime and they suggested it might originate from the nonprognostic lake surface temperatures. A prognostic lake model was included in the model version used in this study, and thus the warm bias might have stemmed from other reasons, such as from SURFEX's own features or the possible biases in ERA-Interim's sea-surface temperatures or sea-ice concentrations that are used to force the sea surface in HCLIM. On the other hand, the HCLIM38-ALARO results for mean seasonal $T_{air}$ were in agreement with EURO-CORDEX RCMs

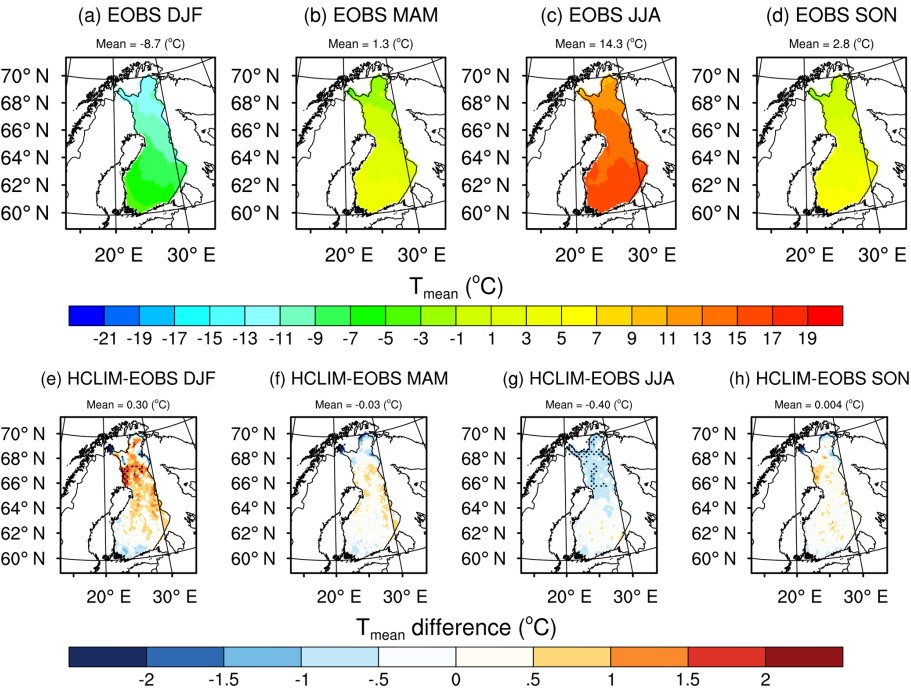

**Figure 3. (a–d)** The reference values of mean near-surface air temperatures ($T_{mean}$) from E-OBS data and **(e–h)** the biases of HCLIM38-ALARO modeled $T_{mean}$ with a reference to E-OBS. The seasonal means were calculated over the whole model domain for the time period of January 2002–December 2014. Stippled areas represent statistically significant differences with $p$ values $< 0.05$.

that were run at 12.5 km grid resolution. For instance, Kotlarski et al. (2014) showed that some of the ERA-Interim-driven EURO-CORDEX RCMs had a warm (cold) bias especially over the northern parts of Finland during the winter (summer). However, a more detailed analysis of the causes of the model biases is out of the scope of this study.

Figure 4 demonstrates that the mean monthly biases in the simulated daily $T_{air}$ with a reference to the E-OBS dataset were generally between $\pm 1\,°C$ when the biases were averaged over different regions of Finland for the period of 2002–2014. The highest positive biases occurred in the winter season and the highest negative biases in the summer as discussed before. However, some regional differences were apparent. For example, in Southern Finland, the biases were mainly negative during the autumn and winter months (October–February). Similarly, the biases were negative at the beginning of the winter season in Western and Central Finland but the biases during the late winter and early spring season were positive as opposed to the negative biases in Southern Finland (excluding March when the bias in Southern Finland was also positive). In Eastern Finland, the mean biases resembled Western and Central Finland but were slightly higher for every month except for July, November, and December. The monthly biases were even higher in Northern Finland and Lapland compared to the other parts of Finland. In the northernmost areas, the biases were mostly positive during the autumn and winter seasons and negative during the spring and summer.

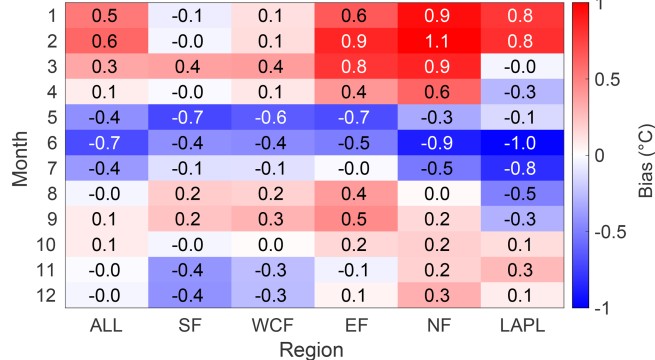

**Figure 4.** The monthly mean biases of simulated near-surface air temperature averaged over Southern Finland (SF), Western and Central Finland (WCF), Eastern Finland (EF), Northern Finland (NF), Lapland (LAPL), and the whole of Finland (ALL) in 2002–2014 with a reference to E-OBS.

### 3.1.2 Minimum and maximum near-surface air temperature and percentiles of mean temperature

Similarly to the mean near-surface $T_{air}$, we assessed the differences between the observed and modeled daily PDFs as well as the multiyear seasonal means of daily minimum and maximum near-surface temperatures ($T_{air,min}$ and $T_{air,max}$, respectively) in 2002–2014 over Finland. Again, the PDFs of both $T_{air,min}$ and $T_{air,max}$ were adequately rep-

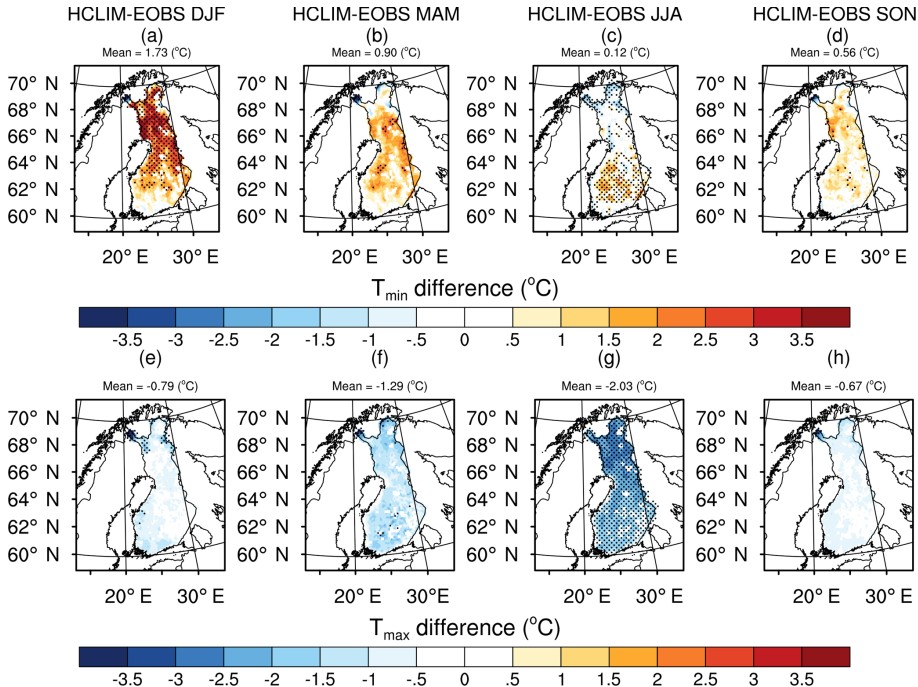

**Figure 5.** The biases in the simulated seasonal means of **(a–d)** minimum near-surface air temperature ($T_{\min}$) and **(e–h)** maximum near-surface air temperature ($T_{\max}$) with a reference to E-OBS. The seasonal mean biases were calculated over Finland for the time period of January 2002–December 2014. Stippled areas represent statistically significant differences with $p$ values $< 0.05$.

resented in HCLIM38-ALARO with the largest deviations in the winter season (not shown). Figure 5 shows that the multiyear seasonal means of $T_{\mathrm{air,min}}$ were mainly overestimated and, contrarily, $T_{\mathrm{air,max}}$ underestimated. The stippled areas in Fig. 5 depict significant differences pointed out by the Student's $t$ test ($p < 0.05$). The differences between HCLIM38-ALARO and E-OBS were significant mainly in the winter and summer season for $T_{\mathrm{air,min}}$ with the largest domain-averaged difference of $1.73\,°C$ found in the winter. For $T_{\mathrm{air,max}}$, the differences were significant mostly in the summer with also the largest domain-averaged difference of $-2.03\,°C$ occurring in the summertime.

In addition to daily minimum and maximum temperatures, the differences in the 5th, 25th, 75th, and 95th percentiles of the daily-mean $T_{\mathrm{air}}$ between the model and observations were computed for different seasons (Fig. S5). The spatial differences for each season and over all the percentiles were similar to each other with generally more positive biases found for the 5th percentile and more negative biases for the 95th percentile (excluding the autumn), which is in line with the results for $T_{\mathrm{air,min}}$ that are overestimated and $T_{\mathrm{air,max}}$ that are underestimated. In the winter, Finland could clearly be divided into two regions as the biases were positive in the northern parts of Finland and negative in the south (excluding the 5th percentile). For all seasons, the maximum biases in the 5th, 25th, and 75th percentiles occurred in the winter with a maximum domain-averaged difference of $4.9\,°C$ for the 5th percentile. For the 95th percentiles, the largest biases

appeared in the summer with a maximum domain-averaged difference of $-2.2\,°C$.

### 3.1.3 Precipitation and wet-day frequency

Also multiyear mean seasonal precipitation sums were reliably simulated by HCLIM38-ALARO although slight overestimation was evident. Figure 6 depicts both observed multiyear mean seasonal precipitation sums from the E-OBS dataset over Finland in 2002–2014 as well as the differences between HCLIM38-ALARO with a reference to E-OBS. Similarly to the figures shown before, the stippled areas represent significant differences confirmed by the Student's $t$ test ($p < 0.05$). Overall, precipitation was overestimated rather than underestimated throughout the year. The biases were the smallest in the winter with a domain-averaged bias of $16.1\,\%$ TS1 and highest in the spring with a domain-averaged bias of $42.2\,\%$. The largest biases in simulated precipitation occurred in the north of Finland, especially over Lapland, where the biases were also statistically significant for every season. The biases were statistically significant over the whole model domain during the spring and summer season. We stress that E-OBS might suffer from undercatch errors during the winter and spring. The larger biases in the northern parts of Finland might again originate from the sparser observation network in the northernmost domain. The results obtained for HCLIM38-ALARO showed similar magnitude and spatial patterns of the precipitation biases

compared to other EURO-CORDEX RCMs that are mainly overestimating seasonal precipitation over Finland during the winter and summer as shown by Kotlarski et al. (2014).

The overall overestimation of spring and summertime precipitation in HCLIM38-ALARO might be due to too frequent low- and moderate-intensity precipitation events as Lindstead et al. (2015) and Lind et al. (2016) pointed out in their studies of HCLIM36 and HCLIM37. Also the wet-day frequency with a $1\,\mathrm{mm\,d^{-1}}$ threshold was slightly overestimated especially during the spring and summer with the highest domain-averaged bias of 4.6 d per season (Fig. S6). Contrarily, HCLIM38-ALARO slightly underestimated wet-day frequency during the winter (excluding the most northern and southern parts of Finland) with the domain-averaged bias of $-0.2$ d per season. In addition, HCLIM38-ALARO slightly overestimated the relative frequency of daily precipitation over Finland for the intensities that were approximately between 10 and $40\,\mathrm{mm\,d^{-1}}$ in the spring season and 10 and $80\,\mathrm{mm\,d^{-1}}$ in the summer reason (Fig. S3). Otherwise, the PDFs of daily precipitation were adequately captured by HCLIM38-ALARO.

Figure 7 further confirms that precipitation was overestimated over different regions of Finland throughout the year. The mean monthly biases between the regions did not substantially differ from each other. However, the biases were the smallest in Northern Finland during the winter (December–March) and in the southern parts of Finland during the other months (April–November). Consistently, the largest biases were found in Lapland. As already seen in Fig. 6, the largest biases appeared during the spring season (especially between April and May) and the second largest biases during the summer and early autumn season (from June to September).

### 3.1.4 Other variables

The modeled seasonal averages of total cloud fraction (clt), $SW_d$, $LW_d$, RH, and WS were compared against the ERA5 reanalysis product over 2002–2014 since these parameters, excluding clt, were used as inputs for RoadSurf together with $T_\mathrm{air}$ and precipitation. Again, the stippled areas in Fig. 8 illustrate significant differences revealed by the Student's $t$ test ($p < 0.05$). Clt was significantly underestimated throughout the year with the highest domain-averaged bias of $-16.1\,\%$ in the winter (Fig. 8a–d). Consequently, $LW_d$ was significantly underestimated during the winter, summer (in the north), and autumn with the largest domain-averaged bias of $-15\,\mathrm{W\,m^{-2}}$ TS2 occurring in the wintertime (Fig. 8e–h). $SW_d$ was, in turn, mostly significantly overestimated, especially during the autumn when the domain-averaged bias was $10.3\,\mathrm{W\,m^{-2}}$ TS3 (Fig. 8i–l). The biases in $SW_d$ during the winter were small as the received actual $SW_d$ is, in general, limited during this time of the year at the high latitudes. However, negative biases in $SW_d$ were found over the southern parts of Finland during the spring and summer, although

the differences were significant only over restricted areas. These results are in agreement with the previous comparison of clt, $LW_d$, and $SW_d$ between HCLIM36-ALARO and ERA-Interim reanalysis product over northern Europe shown by Lindstedt et al. (2015).

In addition, RH was underestimated in the winter and autumn with a domain-averaged bias of $-4.3\,\%$ during the winter and overestimated during the summer with a domain-averaged bias of $6.3\,\%$ (not shown). WS was mainly underestimated during all seasons with the largest domain-averaged negative bias of $-0.6\,\mathrm{m\,s^{-1}}$ appearing in the winter and autumn seasons (not shown).

### 3.2 Evaluation of HCLIM-driven RoadSurf

#### 3.2.1 Road surface temperature

The meteorological data from HCLIM38-ALARO were used as an input to RoadSurf that was further evaluated against 25 road weather stations in Finland. Here, we mostly concentrate on the evaluation of road surface temperature as it is the main output of RoadSurf. Only the results obtained for an extended winter season from October to April were explored because this period is the most relevant for road maintenance (e.g. salting of the roads and snow plowing) and road safety in Finland. Road surface temperature produced by RoadSurf was evaluated against the observations by calculating the PDFs of observed and modeled daily $T_\mathrm{road}$ at the road weather stations as well as computing mean monthly biases and mean absolute errors (MAEs) using the average monthly road surface temperature values. It is good to keep in mind that the hourly and daily temporal resolutions are the most crucial for road weather because the accident rates might increase rapidly in the case of a sudden change of the prevailing weather (Juga et al., 2012). The monthly timescale was selected for the evaluation to account for the fact that RoadSurf was driven using an RCM that was forced by a reanalysis product only in the lateral boundaries. This implies that the modeled day-to-day variability might not entirely match with observations at all locations. However, calculating monthly statistics gives us a clear understanding of the model performance for different months during the study period from 2002 to 2014.

Figure 9 makes it evident that the HCLIM-driven RoadSurf was able to simulate the monthly means of $T_\mathrm{road}$ with high accuracy and with most of the biases falling between $\pm 2\,^\circ\mathrm{C}$. The mean monthly bias at all 25 stations was $-0.3\,^\circ\mathrm{C}$ (range $-2.1$ to $2.7\,^\circ\mathrm{C}$) and MAE $0.9\,^\circ\mathrm{C}$ (range 0.3–$2.9\,^\circ\mathrm{C}$). Some regional and seasonal differences were apparent. In January and February, most of the stations located in Southern Finland and Western and Central Finland had mainly negative mean biases, whereas the biases were predominantly positive at the stations located in Eastern Finland, Northern Finland, and Lapland. When looking at the results for all stations, most of the positive mean biases oc-

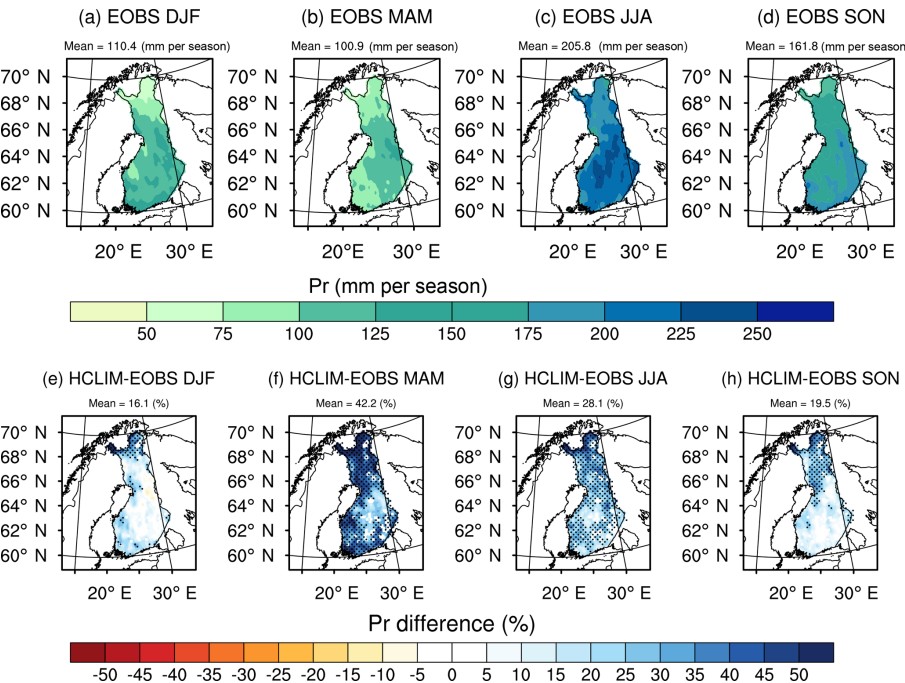

**Figure 6. (a–d)** The reference values of precipitation (Pr) from E-OBS data and **(e–h)** the biases of HCLIM38-ALARO modeled Pr with a reference to E-OBS. The seasonal averages were calculated for the time period of January 2002–December 2014. Stippled areas represent statistically significant differences with $p$ values < 0.05.

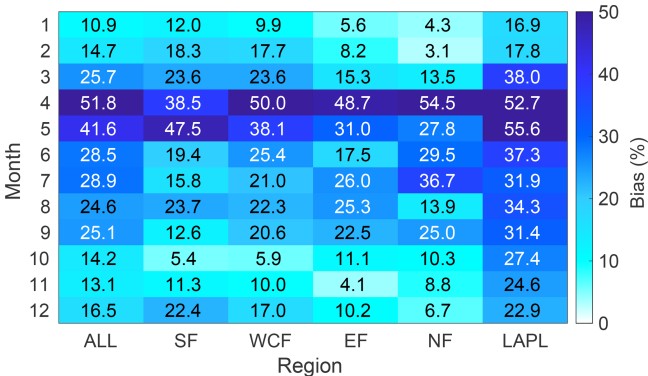

**Figure 7.** The monthly mean biases of simulated precipitation averaged over Southern Finland (SF), Western and Central Finland (WCF), Eastern Finland (EF), Northern Finland (NF), Lapland (LAPL), and the whole of Finland (ALL) in 2002–2014 with a reference to E-OBS.

curred in January and March, whereas negative biases occurred in April, November, and December. Eleven stations had negative mean biases throughout all the analyzed months while the rest of the stations had both negative and positive mean biases depending on the month. Overall, the MAE values were the lowest in March and October while the highest MAE values occurred in Lapland in January and February. Despite the apparent mean monthly biases, the shapes of the daily $T_{\mathrm{road}}$ PDFs were sufficiently reproduced by RoadSurf

with the largest deviations found in the winter (Fig. S4) in accordance with the PDFs of daily $T_{\mathrm{air}}$.

Probable reasons for the seasonal and regional differences in the model performance are the biases in the HCLIM38-ALARO data and the fact that RoadSurf works well in the vicinity of 0 °C. To address the impact of the biases in the input parameters on the $T_{\mathrm{road}}$ biases, we computed the monthly mean biases in the HCLIM38-ALARO model outputs with a reference to E-OBS (in the case of $T_{\mathrm{air}}$ and precipitation) and ERA5 (in the case of $LW_{\mathrm{d}}$, $SW_{\mathrm{d}}$, RH, and WS) at the grid cell closest to the road weather station in question. The monthly biases in the input parameters were plotted against the monthly biases in $T_{\mathrm{road}}$. The analysis shown in Fig. 10 revealed that the variability in the monthly biases of $T_{\mathrm{air}}$ explained on average 57 % (range 19 %–84 % in October–April) of the variability in the monthly biases of $T_{\mathrm{road}}$ while the $LW_{\mathrm{d}}$ biases explained on average 16 % (range 2 %–34 % in October–March). Furthermore, the variability in $SW_{\mathrm{d}}$ biases was found to explain a small amount (4 %) of the variability in $T_{\mathrm{road}}$ biases during April. The comparison between other input parameters and $T_{\mathrm{road}}$ did not reveal clear linear responses and are thus not discussed here. Also, Karsisto et al. (2017) noted that a part of the $T_{\mathrm{road}}$ biases is caused by the biases in the input parameters used to force road weather models. In their study, the input was provided by a forecast produced with a high-resolution NWP version of HARMONIE (cy36h1.4) with a grid resolution of 2.5 km over the Netherlands. In that study, the KNMI (the Royal Netherlands

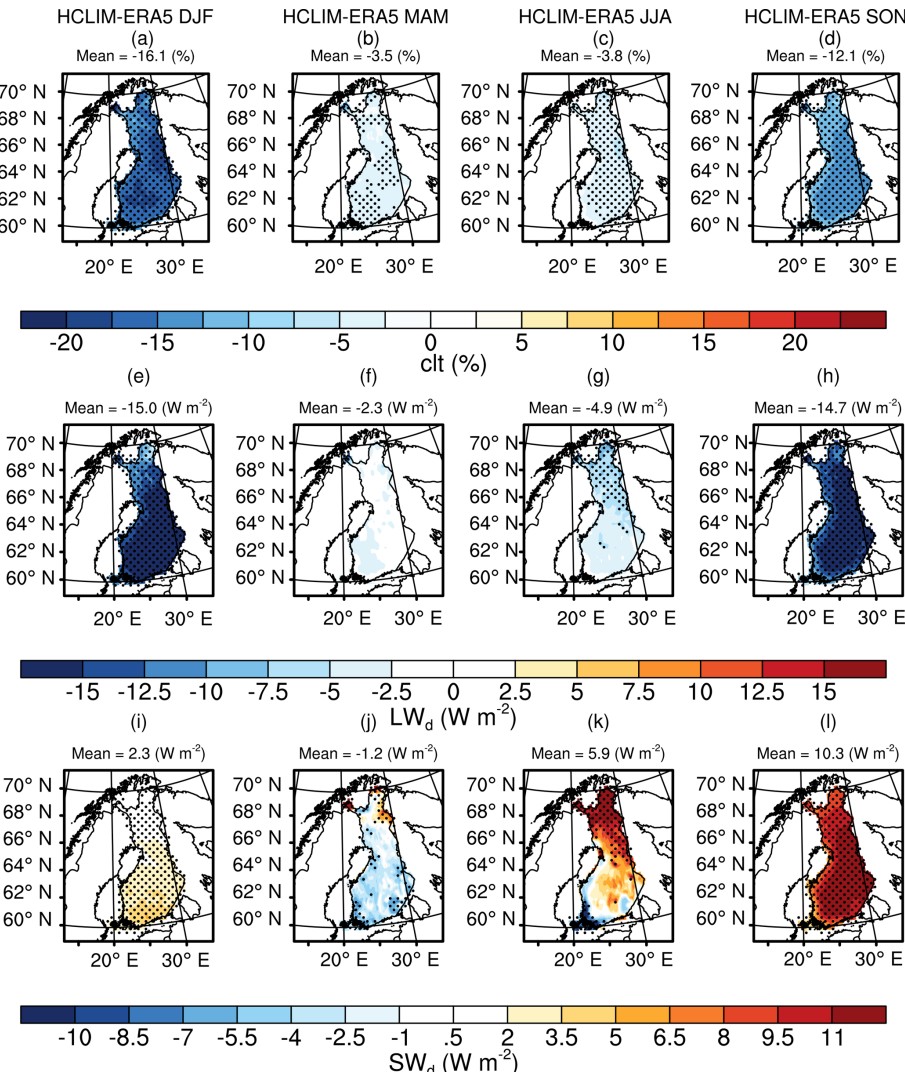

**Figure 8.** TS4 The biases in the simulated seasonal means of **(a–d)** total cloud fraction (clt), **(e–h)** downwelling longwave and **(i–l)** shortwave radiation (LW$_d$ and SW$_d$, respectively) with a reference to the ERA5 reanalysis product. The seasonal mean biases were calculated over Finland for the time period of January 2002–December 2014. Stippled areas represent statistically significant differences with $p$ values < 0.05.

Meteorological Institute) road weather model (a 1-D heat balance model similar to RoadSurf) was run by removing the bias of one of the model inputs, 2 m $T_{air}$. This reduced the $T_{road}$ bias during the nighttime by 50 % indicating that the biases in the input parameters clearly affect road weather model outcomes.

Moreover, the comparison of the simulated and observed $T_{air}$ in the wintertime (December–February) revealed a warm bias ranging from 0.1 to 1.1 °C in the northern parts of Finland (Northern Finland and Lapland) while Southern Finland had negative biases ranging between −0.4 and −0.04 °C (see Fig. 4). Thus, the larger and more positive biases in the simulated $T_{air}$ in Northern Finland and Lapland compared to Southern Finland seem to explain the larger positive biases in the modeled $T_{road}$ at the northernmost stations. In addition, Kangas et al. (2015) noted that RoadSurf is designed to work especially well when temperatures are close to 0 °C. Based on the monthly statistics obtained for the study period (2002–2014), road surface temperatures were crossing 0 °C particularly often during March, April, and October (see Sect. 3.2.2). This good model performance near 0 °C could, in turn, partly explain why the MAE values were lower in October and March compared to other months.

Some part of the biases in $T_{road}$ might originate from the RoadSurf model itself. For instance, the absence of snow removal and salting in the model might keep the road surface colder than what it would be with the maintenance actions. In addition, traffic is assumed to pack some part of the snow into ice while the remaining part is assumed to be blown away from the road. For example, the real traffic amounts

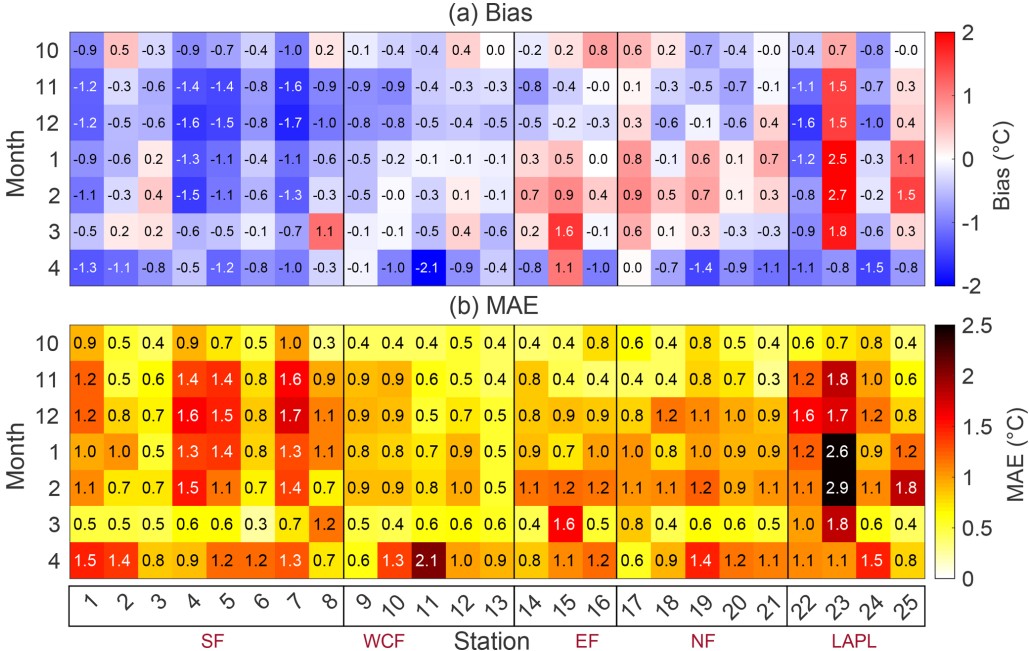

**Figure 9. (a)** The mean monthly biases and **(b)** MAE values of simulated road surface temperature between October and April in 2002–2014. The station indices on the $x$ axis refer to Table 1. SF refers to Southern Finland, WCF to Western and Central Finland, EF to Eastern Finland, NF to Northern Finland, and LAPL to Lapland.

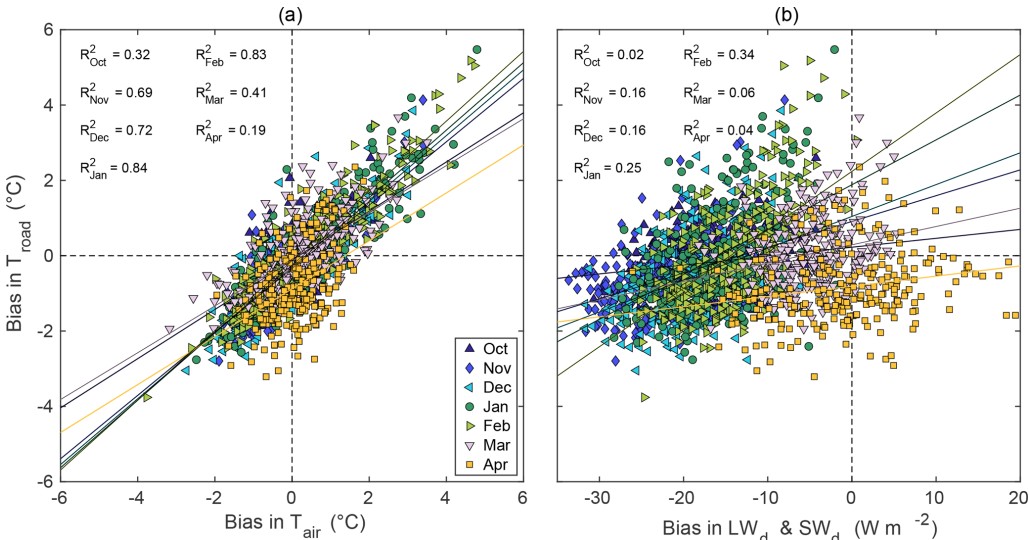

**Figure 10.** Scatter plots of the mean monthly biases of road surface temperature ($T_{road}$) against **(a)** the mean monthly biases of near-surface air temperature ($T_{air}$) and **(b)** the mean monthly biases of downwelling longwave (LW$_d$ for October–March) and shortwave radiation (SW$_d$ for April) at the road weather stations. The squared $R$ values represent linear regression for different months with $p$ values $< 0.001$ ($p$ value for LW$_d$ in October 0.01).

are higher in Southern Finland compared to the other parts of the country, which can lead to an overestimation of the simulated icy and snowy conditions in the south and, hence, to colder road surface conditions than what is observed. On the other hand, the snowpack that is observed might actually stay longer than what is simulated by the model leading

to positive biases in $T_{road}$ at locations with less traffic: this could especially happen at stations such as station 23 (Sieppijärvi). The biases in $T_{road}$ might also stem from the absence of shading effects as this effect is not taken into account by RoadSurf.

Although the results obtained in this study indicated a good skill of RoadSurf to realistically capture $T_{road}$, the mean biases were slightly larger compared to the previous studies of RoadSurf. For example, Karsisto et al. (2016) found that the biases in the simulated $T_{road}$ varied between $-1$ and $1\,°C$ (mostly $\pm 2\,°C$ in our study) at 20 stations in Finland during October and December 2013 when RoadSurf was driven by a high-resolution NWP version of HARMONIE (cy36h1.4) with a grid resolution of 2.5 km without any data assimilation. However, it is good to note that the results obtained in our study and by Karsisto et al. (2016) are not directly comparable since in their study RoadSurf was initialized using road weather station measurements for 48 h and only the first forecasted hour was analyzed. However, one possible reason for the slightly larger errors obtained in the present study might be the coarser grid resolution of HCLIM38-ALARO as compared to the NWP version: coarser grid resolution implies that not all the local features, such as topography, are described as in detail as they are in higher resolution NWP models. Increasing the grid resolution of HCLIM38-ALARO might therefore yield better performance for RoadSurf although increasing the grid resolution of a climate model will also increase the computational cost. However, the longer time period used in this study makes the results more robust compared to the previous studies in which only short time periods were analyzed.

### 3.2.2 Zero-crossing days

Temperatures close to $0\,°C$ should be predicted correctly because in these conditions wet road surfaces have a tendency to freeze (e.g. Vajda et al., 2014) and roads are the most slippery in the copresence of ice (Moore, 1975). In this study, a zero-crossing day was defined as a day when the road surface temperature had been at least once both below $-0.5\,°C$ and above $0.5\,°C$.

Figure 11 shows that the monthly number of zero-crossing days and the monthly variation (standard deviation) were well captured by RoadSurf. This was expected as RoadSurf has been confirmed to simulate $T_{road}$ accurately in the vicinity of $0\,°C$ (Kangas et al., 2015; Karsisto et al., 2016). On average, the correlation coefficient was very high (0.92) and the mean bias was approximately 0.9 d (Fig. 11f). The performance of the model differed slightly depending on the analyzed region. Surprisingly, the correlation coefficient was the lowest in Southern Finland and the highest in Northern Finland and Lapland, whereas the bias was the lowest in Eastern Finland and the highest in Lapland. The higher biases in Lapland might be explained by the overall overestimation of zero-crossing days, which might, in turn, be caused by the warm bias in the simulated $T_{road}$ values as discussed before. Overall, most of the zero-crossing days occurred in March, April, and October. However, the number of zero-crossing days declined in March and increased in April when moving towards the north. In Lapland, most of the zero crossings

occurred in April instead of March. This was also expected as the winter season (and therefore the coldest period) lasts longer in Lapland compared to the southern parts of Finland, leading to less zero-crossing days in March. The smallest number of zero crossings took place in January, February, and December. These are usually the coldest months of the year, especially in Lapland (see also Table 1); thus, $0\,°C$ is not crossed as often during these months.

### 3.2.3 Road surface classes

The majority of the wintertime and weather-related road accidents in Fennoscandia are caused by the snowy and icy road conditions in addition to, for example, driving habits and worn out tires (Salli et al., 2008). To investigate RoadSurf's skill to correctly predict the road surface classes (e.g. snowy and icy surfaces), the model results and observations were compared by calculating the fraction of each road surface class occurring within a month. The fraction was calculated as a multiyear sum of the occurrence of the surface class in question divided by the multiyear sum of the occurrence of all surface classes and then taking an average between stations falling into the same region. It is good to remember that the observed and modeled road surface classes might not fully match as they are defined differently.

Figure 12 shows that overall RoadSurf captured well the prevailing road surface conditions although the observed and modeled fractions differed slightly. For example, the model overestimated the fraction of dry surfaces in all regions (average bias over all regions and all months was 7 % as a fraction) and underestimated damp surfaces slightly more (average bias $-16$ %). The model underestimated also wet surfaces (average bias $-6$ %), but the fraction of the partly icy class (8 % on average) was almost equal to this difference between the modeled and observed wet surface fraction. Therefore, these results indicated that wet surfaces tended to be predicted as partly icy, although it has to be remembered that observations do not have a partly icy class. The underestimation of the frost on the road (average bias $-1$ %) and overestimation of ice (2 %) were also of a similar magnitude with opposite signs. Moreover, the snow class was slightly overestimated with an average bias of 2 %. These results are in line with the study by Kangas et al. (2015) where they encountered an overestimation of ice and snow storages produced by RoadSurf at two stations located in Finland. In addition, they found that sometimes frost predicted by the model was observed as ice in the measurements. In the present study, frosty surfaces were, however, mainly underestimated. On the other hand, both icy and frosty surfaces are slippery, so in that aspect the model behavior (i.e., the tendency of the model to underestimate frost and to overestimate ice with the same magnitude) is acceptable.

The absence of road maintenance could be one logical reason why the model overestimated icy and snowy surfaces: in real life, salting prevents roads becoming icy and snow is re-

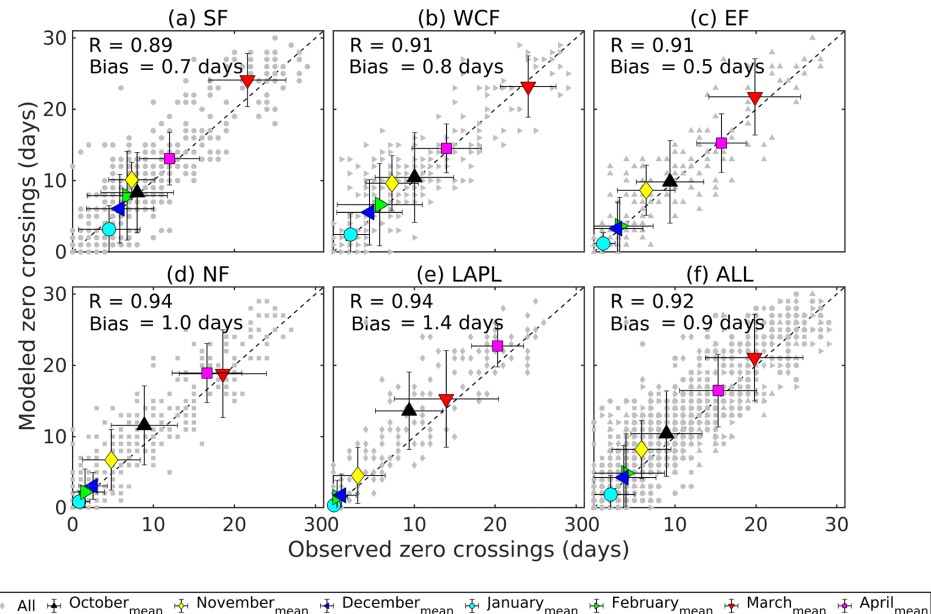

**Figure 11.** Modeled vs. observed days per month when road temperatures had been both below −0.5 °C and above 0.5 °C (zero-crossing day) during October and April in 2002–2014 in **(a)** Southern Finland (SF), **(b)** Western and Central Finland (WCF), **(c)** Eastern Finland (EF), **(d)** Northern Finland (NF), **(e)** Lapland (LAPL), and **(f)** the whole of Finland (ALL). Grey color represents the monthly values for every year and the multiyear monthly means are illustrated in other colors. The vertical and horizontal bars represent ±1 standard deviation based on 13 years of monthly values from the model and observations, respectively. $R$ stands for the Pearson correlation coefficient and BIAS for the mean difference between the modeled and observed values. The dashed black line represents a 1 : 1 reference line.

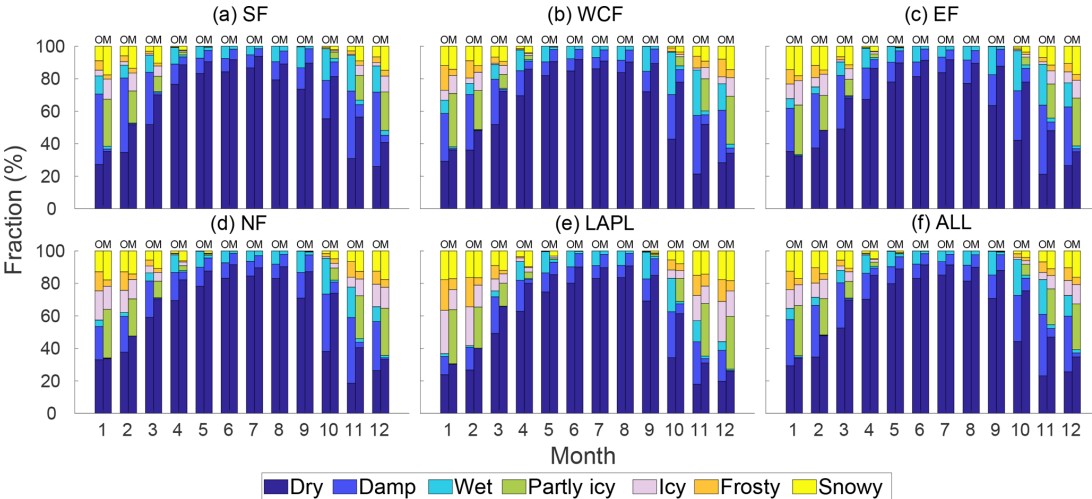

**Figure 12.** Observed (O) and modeled (M) fractions of road surface classes (e.g. dry, wet, or icy) within each month in 2002–2014 in **(a)** Southern Finland (SF), **(b)** Western and Central Finland (WCF), **(c)** Eastern Finland (EF), **(d)** Northern Finland (NF), **(e)** Lapland (LAPL), and **(f)** the averages for whole of Finland (ALL). The definitions of road surface classes differ slightly for the observations and model (e.g. the partly icy class is included only in the model).

moved from the roads. Accordingly, the observed and modeled fractions of snowy surfaces were very similar to each other in Lapland where maintenance, such as snow plowing, is performed far less frequently compared to the more southern parts of Finland in real life. The icy road fraction was underestimated in Lapland, whereas this fraction was overestimated in the other regions: in reality, salting is not performed as often at the stations in Lapland as in Southern Finland and thus icy roads can occur more frequently in the northernmost stations. Furthermore, the RoadSurf model takes the effect of traffic into account in a similar manner regardless of the region. Therefore, the simulated ice and snow

storages might deplete too fast in the model considering the substantially lower traffic amounts in the northern parts of Finland compared to the south. For instance, snow storage was slightly underestimated in Lapland although only in Jan-
5 uary and November (Fig. 12e). The warm bias in Lapland might also have played a role in the underestimation of icy road fraction as icy roads are less likely to occur if the simulated air temperatures are too high. In addition, the underestimated wet and damp surfaces during the winter months
(December–February) might be explained by the slightly underestimated wet-day frequency of precipitation over most parts of Finland (see Fig. S6).

### 3.2.4 Categorical performance of the simulated frequency of water, snow, and ice storages

Rainfall is considered to be one of the main contributing factors in traffic accidents together with snow and ice on the road (e.g. Andersson and Chapman, 2011b). Therefore, the water, snow, and ice storages, as well as their frequency, should be simulated accurately. The absolute values of the
storages are not discussed here as the modeled values represent areal averages and observations represent point measurements. In addition, the optical sensor might not correctly sense the exact thickness of the water, snow, or ice layer on the road but rather it might detect only the upper layer of
these storage terms. Thus, RoadSurf's ability to simulate the frequency of the storages was assessed by first calculating the daily maximum values of the storages between October and April and, further, setting the daily values to one if the daily maximum value was more than zero and to zero if the daily
maximum value was zero. These binary values were used to calculate hits and false alarms (Table S1 in the Supplement) and the probability of detection (POD) and false alarm ratios (FARs) (Roebber, 2009). The details of the POD-FAR analysis are explained in the Supplement Sect. S1. The num-
ber of compared daily cases per station varied between 503 and 1101 d depending on the data availability at each station. However, this method might penalize the model more than it should because the modeled storages were compared with observations using day-to-day values. For this reason,
we additionally calculated the multiyear sums of all the modeled and observed daily cases with daily maximum more than zero or zero.

The results of the POD-FAR analysis for 11 stations including an optical sensor (see Table 1) are illustrated in
Fig. 13 using a categorical performance diagram (Roebber, 2009; please see the Supplement Sect. S1 for more details). Figure 13 shows that RoadSurf reliably captured the occurrence of the storage terms as the points located near the upper-right corner of the diagram. However, the model per-
formance varied slightly depending on which storage was simulated. For instance, the modeled water storages had the lowest FAR (highest 1–FAR) values but also the lowest POD values. This means that because the model did not detect

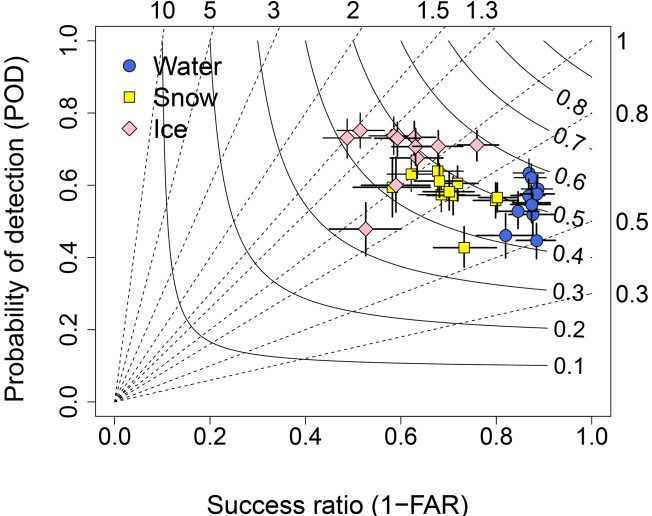

**Figure 13.** The performance diagram of water, snow, and ice storages modeled at the 11 road weather stations which have an optical sensor (see Table 1). Absolute values of the modeled and observed maximum daily storages were not used directly; but instead, the daily value was set to one if the maximum value was more than zero and to zero if the maximum value was zero. The months between October and April were included in the analysis. Success ratio (1–FAR) runs along the $x$ axis and POD along the $y$ axis. Dashed lines represent the frequency bias and continuous lines the CSI (critical success index, see the Supplement Sect. S1). The vertical and horizontal lines represent the 95 % confidence intervals for POD and FAR values, respectively, calculated by using a bootstrap method and 1000 resamples.

water as often as it should, the false alarm ratio was also smaller. The frequency bias values were lower than one indi-55 cating an underestimation of the events with water on the surface. The opposite was true for the modeled ice storages: the events were predicted well (POD was high) but false alarms were more frequent (1–FAR was lower). Furthermore, the frequency bias values were greater than one suggesting an 60 overestimation of the events with ice on the road. The POD and FAR values of the modeled snow storages fell somewhere between the POD and FAR values which were obtained for the water and ice storages. The model underestimated the frequency of the events with snow on the road but 65 to a lesser extent compared to the underestimated frequency of the water storages.

It has to be emphasized once more that the model does not take into account road maintenance measures. Again, the absence of salting can be one reason for the overestimated oc-70 currence of ice and the underestimated occurrence of water on the road surface. However, the model is thus on the "safe side", which means that in operational use the model would give warnings to the road users slightly more often than what would be required. As mentioned before, a part of the un-75 derestimated frequency of water might be explained by the slightly underestimated wet-day frequency of precipitation

during the winter season. On the other hand, the absence of snow removal in the model did not lead to an overestimated frequency of snow on the road: this frequency was underestimated while the fraction of snowy road cover was overestimated as shown in Fig. 12. One possible reason for this discrepancy might be the different number of road weather stations used in the POD and FAR analysis compared to the road condition analysis (11 vs. 25 stations). Another reason might be that the POD and FAR analysis utilized fewer observations compared to the number of observations used in the analysis of the road surface conditions (due to a higher amount of missing data). Moreover, the RoadSurf-HCLIM configuration might not capture all the snow events which are observed at the station because the simulated storages represent areal averages. However, the underestimated frequency of snow cannot be explained by the snowpacks that are depleting too fast in the model. This is because the majority of the stations with an optical sensor utilized in this study are located in the southern parts of Finland where the modeled snowpacks might actually stay longer compared to the measurements as discussed before.

In addition to the POD-FAR analysis, we computed the modeled and observed fractions of the multiyear sums of the daily cases with the daily maximum storage of water, snow, or ice more than zero or zero. The results are shown in Fig. S7 as fractions over all 11 stations. This figure supports the main conclusions from the POD–FAR analysis: the occurrence of water and snow storages were underestimated by the model by $-18\%$ and $-7\%$, respectively. The frequency of ice storage was slightly overestimated by $5\%$.

## 4 Conclusions

This study described the performance of the HCLIM38-ALARO regional climate model over Finland and, further, evaluated the skill of the HCLIM38-ALARO-driven road weather model RoadSurf to reproduce the present-day road weather conditions in Finland. HCLIM38-ALARO was forced with the reanalysis product ERA-Interim in the lateral boundaries. This study showed that HCLIM38-ALARO is in good agreement with the gridded daily mean air temperature and precipitation observations: the model reliably reproduced the monthly and seasonal temporal and spatial patterns as well as daily variability in these variables over Finland. Especially daily-mean near-surface air temperatures were well represented by HCLIM38-ALARO. On the other hand, daily minimum air temperatures were slightly overestimated and daily maximum temperatures underestimated. Precipitation was overestimated during all seasons, although some of this overestimation might be caused by the inaccuracy of E-OBS data due to possible undercatch errors and lower station density in the northern parts of Finland. Overall, the HCLIM38-ALARO results were found to be in line with other EURO-CORDEX RCMs. The underestimated to-

tal cloud fraction in the model led to the overestimated downwelling shortwave and underestimated longwave radiation, which has also been encountered in the previous evaluations of HCLIM over northern Europe in the wintertime.

As far as the authors are aware, this may be the first paper that studies the performance of a road weather model which is forced by RCM data. This study revealed that the HCLIM38-ALARO-driven RoadSurf was able to adequately reproduce the daily distributions of road surface temperatures ($T_{road}$) and accurately simulate $T_{road}$ with the mean monthly bias of $-0.3\,°C$ and the mean monthly MAE of $0.9\,°C$ over Finland. These metrics indicated a slightly poorer performance than what was obtained in the earlier studies of RoadSurf. The coarser grid resolution of the HCLIM38-ALARO compared to the NWP model input used in the earlier studies might be the main reason for this outcome as no data assimilation was used for HCLIM38-ALARO or the NWP model. Moreover, the HCLIM38-ALARO simulated air temperature tended to have a warm bias over the northern parts of Finland in the winter. This, in turn, might be the major reason for the better performance of RoadSurf to simulate $T_{road}$ at the stations located in the southern parts of Finland compared to the stations located in Lapland. The variability in the air temperature biases was found to explain the largest part of the variance in the road surface temperature biases as compared to other input variables of RoadSurf.

In addition, RoadSurf adequately captured the daily zero crossings, which verified the good performance of the model when temperatures approach $0\,°C$. This is of great importance as the road surfaces are most prone to slippery conditions when the road surface temperatures are close to $0\,°C$ and simultaneous icing occurs. Moreover, the analysis on the road surface classes showed that the model is overall in a good agreement with the observations in terms of the prevailing road conditions. However, the model tended to yield more icy and snowy road surfaces than indicated by observations. The absence of road maintenance, such as salting and snow plowing, is very likely the dominant reason for this model behavior as well as for the overestimated occurrence of ice and underestimated occurrence of water on the road surface. On the other hand, the overestimated traffic wear in the model and therefore a depletion of ice storages that is too fast could be the reason for the underestimated fraction of icy surfaces at the northernmost stations.

These results were obtained using a limited set of road weather stations in Finland. On the other hand, the 13-year-long study period makes the results more robust compared to the earlier studies of RoadSurf which have concentrated only on short verification periods of 1 week to some months. Therefore, the results represented in this study indicated that HCLIM38-ALARO realistically captured the climate over Finland and that this RCM data can be used as an input to RoadSurf in order to produce reliable results of $T_{road}$, road surface classes, and storage terms. Although RoadSurf represents a scenario wherein nothing is done in terms of road

maintenance, it also means that the model is ideal to study the relative changes in the road surface conditions due to climate change. Earlier studies of climate change impacts on road weather have mainly considered the relative changes in air temperature and precipitation. Therefore, the approach presented in this study offers an alternative to these methods: running the road weather model with HCLIM38-ALARO produced climate projections makes it possible to directly study how the road weather conditions are going to change in the future.

*Code availability.* The ALADIN and HIRLAM consortia cooperate on the development of a shared system of model codes. The HCLIM model configuration forms part of this shared ALADIN-HIRLAM system. According to the ALADIN-HIRLAM collaboration agreement, all members of the ALADIN and HIRLAM consortia are allowed to license the shared ALADIN-HIRLAM codes within their home country for noncommercial research. Access to the HCLIM codes can be obtained by contacting one of the member institutes of the HIRLAM consortium (see links at http://www.hirlam.org/index.php/hirlam-programme-53, last access: 29 July 2019). Access will be subject to signing a standardized ALADIN-HIRLAM license agreement (http://www.hirlam.org/index.php/hirlam-programme-53/access-to-the-models, last access: 29 July 2019). The RoadSurf code is not in the public domain and cannot be distributed.

*Data availability.* Due to the very large size of the data files, the data are not publicly available. The data files can be requested from the first author.

## Appendix A: The maintenance classes of the roads during wintertime in Finland (Finnish Transport Agency, 2018)

### A1    Maintenance class 1 (lse):

The road is kept bare most of the time. The slipperiness of the roads is prevented beforehand, but mild slipperiness might occur in the case of a rapid change in the prevailing weather. Salting is not possible during long-lasting cold periods, which can lead to partially frozen road surfaces. The maintenance is timed so that inconvenience for traffic is minimized.

### A2    Maintenance class 2 (ls):

The road is kept bare most of the time. The aim is to prevent slipperiness beforehand, but mild slipperiness might occur in the case of a rapid change in the prevailing weather. The central and northern parts of Finland, and also the southern part of the country (only during the coldest periods), might have a thin ridge of snow packed on the road, which does not particularly affect driving. Salting is not possible during long-lasting cold periods, which can lead to partially frozen road surfaces.

### A3    Maintenance class 3 (lb):

The road is kept bare most of the time. The aim is to prevent slipperiness beforehand, but mild slipperiness might occur in the case of a rapid change in the prevailing weather. During the coldest periods, there might be shallow and narrow ridges of snow packed on the road. Salting is not possible during long-lasting cold periods, which can lead to partially frozen road surfaces.

### A4    Maintenance class 4 (l):

The road is maintained at a fairly high standard but mostly without salt. The surface of the road is partially bare depending on the traffic volume and weather. There might be ridges of snow packed on the road and the road might also be fully covered with a snowpack. The road is kept safe enough for the road users. The possible snowpack on the road surface is smoothed. Slipperiness is prevented beforehand only in the autumn and spring and in the case of particularly hazardous situations.

*Supplement.* The supplement related to this article is available online at: https://doi.org/10.5194/gmd-12-1-2019-supplement.

*Author contributions.* ET performed the HCLIM38-ALARO simulations with the help of JPP. JPP performed the offline coupling of RoadSurf and HCLIM38-ALARO. ET planned and performed the analysis of HCLIM38-ALARO and HCLIM-driven RoadSurf with help provided by JPP, MH, HK, and AL. MK and MH assisted with the road weather model RoadSurf and MH with the road weather observations. JPP, HK, and AL initiated the work. ET wrote the paper. All co-authors participated in the paper writing phase and gave valuable comments regarding the first versions of the article.

*Competing interests.* The authors declare that they have no conflict of interest.

*Acknowledgements.* The leading author would like to thank Maj and Tor Nessling foundation for the financial support of regional climate modeling research. We also acknowledge the E-OBS dataset from the EU-FP6 project UERRA (http://www.uerra.eu, last access: 29 July 2019), the data providers in the ECA&D project (http://www.ecad.eu, last access: 29 July 2019) as well as the ERA5 reanalysis product from ECMWF. We are grateful for the help provided by Ari Aaltonen in retrieving the Finnish road weather observations. In addition, we would like to thank the two anonymous reviewers for their comments and efforts towards improving our article.

*Financial support.* This research has been supported by the H2020 European Research Council (ECLAIR (grant no. 646857)), the Academy of Finland (grant no. 287440), and the Maj and Tor Nessling foundation.

*Review statement.* This paper was edited by David Lawrence and reviewed by two anonymous referees.

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

## Remarks from the typesetter