# Peer review of "The road weather model RoadSurf (v6.60b) driven by the regional climate model HCLIM38: evaluation over Finland"

_Geoscientific Model Development, 2018_

## Short Comment (SC1) · 6 Mar 2019

Dear authors,

In my role as Executive editor of GMD, I would like to bring to your attention our Editorial version 1.1:

http://www.geosci-model-dev.net/8/3487/2015/gmd-8-3487-2015.html

This highlights some requirements of papers published in GMD, which is also available on the GMD website in the 'Manuscript Types' section:

http://www.geoscientific-model-development.net/submission/manuscript_types.html

[Figure]

In particular, please note that for your paper, the following requirements have not been met in the Discussions paper:

- "The main paper must give the model name and version number (or other unique identifier) in the title."

- "All papers must include a section, at the end of the paper, entitled 'Code availability'. Here, either instructions for obtaining the code, or the reasons why the code is not available should be clearly stated. It is preferred for the code to be uploaded as a supplement or to be made available at a data repository with an associated DOI (digital object identifier) for the exact model version described in the paper. Alternatively, for established models, there may be an existing means of accessing the code through a particular system. In this case, there must exist a means of permanently accessing the precise model version described in the paper. In some cases, authors may prefer to put models on their own website, or to act as a point of contact for obtaining the code. Given the impermanence of websites and email addresses, this is not encouraged, and authors should consider improving the availability with a more permanent arrangement. After the paper is accepted the model archive should be updated to include a link to the GMD paper."

As evaluations are also model version specific, please add the version number of the evaluated RoadSurf model to the title of the manuscript upon revision for GMD.

Furthermore, it is not sufficient to state that the code you are evaluating is not available. At least state, why the code is not available. If there are no license reasons to prevent the publication the code should be made publicly available.

Yours,

Astrid Kerkweg

---

## Referee Comment (RC1) · Anonymous Referee #1 · 15 Mar 2019

MS No.: gmd-2018-330
Review, 11 March 2019

Title: The road weather mode RoadSurf driven by the HARMONIE-Climate regional climate model: evaluation over Finland

Authors: Erika Toivonen, Marjo Hippi, Hannele Korhonen, Ari Laaksonen, Markku Kangas, and Joni-Pekka Pietikäinen

Recommendation: [Major Revisions]

GENERAL COMMENTS:

This article presents an evaluation study across Finland of a 13-year long record of road surface temperatures and road ice, snow, and water storage parameters obtained with a road surface weather model driven by output from a regional climate model operated at 12.5 km resolution. The RCM is in turn forced at its lateral boundaries by atmospheric information from the ERA-Interim re-analysis. The emphasis of the analysis is on the performance of the road surface model compared with observations obtained at 25 road weather stations of which 11 were also equipped with optical sensors to establish the prevailing condition of the road surface.

Overall, the paper is coherently written, but in my opinion the scope is too much from the perspective of an NWP. The entire analysis assumes as if the modelling chain can be one-to-one compared with observations and a statistical machinery is applied resulting in skill scores which one usually sees in assessing the forecast performance of a prediction model. This approach does not match with the purpose of the study to evaluate a model system when operated in climate mode but still using observations (ERA-Interim) to constrain the large-scale model circulation to the observed synoptic-scale structure.

The next step, also mentioned by the authors, will be to run the RCM-RoadSurf modelling system driven by GCM output resulting from transient multi-annual simulation under prescribed emission scenarios. The biases then found will presumably be much larger than seen in this evaluation study, and any performance rating as if it were a prediction model will be deemed meaningless. The primary reason for that is huge biases in circulation and regime statistics in the GCM drivers compared to ERA-Interim. So, the authors can better focus on the role of circulation and regime drivers on the performance of their modelling chain, than focus on skill scores like RMSE and Pearson's correlation coefficients. Eventually, they want to draw credible conclusions how climate change information at the large scale will propagate through their RCM to the RoadSurf model.

In addition I would argue that the way the experiment has been set up makes it very difficult to conclude how the shortcomings in performance can be attributed to the model components that are used. Several times the authors mention that an issue might be related either to the warm and/or wet bias in their RCM or to features in the RoadSurf model. In that respect, I am wondering why the authors have not carried out a bias

adjustment to HCLIM-ALARO temperature and precipitation which serve as forcings to the RoadSurf model. Such an additional experiment would have a twofold benefit: a) to disentangle the bias in HCLIM from issues in RoadSurf, and b) to obtain a measure to what extent the biases in HCLIM affect the performance of RoadSurf. The latter would be very helpful in the analysis and interpretation of future GCM-driven experiments.

MAJOR COMMENTS:

1) I would strongly suggest to focus on the Finland area from the beginning. The discussion of the HCLIM-ALARO model performance for the whole of the Fenno-Scandian domain is distracting. There are always huge issues in the mountainous areas in Norway, for any RCM, and also in E-OBS, but they are not relevant for this study. Focus on Finland in Figs 3 and 5.

2) Do not only examine the bias in the monthly mean temperature, but also at a number of percentiles (e.g. P5,25,75,95). The diurnal amplitude in model temperature compared to observations is relevant here as well.

3) Similarly for precipitation. In addition to mean precipitation look at wet-day frequency (threshold 0.3 or 1.0 mm/day), and perhaps some exceedance percentiles. It provides much more insight than an RMSE score.

4) Can there be said anything about the accuracy of the RCM inputs other than near-surface temperature and precipitation that are used to drive the RoadSurf model.

5) As mentioned in the general comments it would be useful to apply a bias-adjustment on daily mean temperature and precipitation, also frequency of occurrence, to bring the HCLIM-ALARO temperature (e.g. quantile-quantile) and precipitation forcing in the same "statistical" ballpark as the observations.

6) As the RCM is operated at 12.5km resolution there should be reference to the efforts within EuroCordex in conducting evaluation (ERA-Interim driven) and transient (GCM driven) experiments at 12.5 km resolution across Europe with a variety of RCMs. For the evaluation study you best cite Kotlarski et al. (2014; doi: 10.5194/gmd-7-1297-2014).

7) Section 3.2.1 ("Road surface temperature"), after line 240 bothers me most. Why are all discrepancies blamed on the bias in temperature forcing, and not on potential issues with downwelling radiation, in particular biases in downwelling long wave radiation due to biases in cloud amount or cloud base.

8) Page 9, L260-266. The authors argue that the better skill obtained with the forcing from the NWP compared to this study can be ascribed to the higher resolution at which the NWP is operated. I tend to disagree on that, in my opinion the use of dataassimilation when operating in NWP-mode will keep the model atmospheric state across the Finland region much closer to the observed state.

9) The statistical methods used in sections 3.2.2. are not suitable for evaluation purposes, they belong to the realm of NWP verification. I advise to take this section out or move it to the supplement.

10) The same applies to section 3.2.5 although I find the message (i.e. over-representing of storage of ice, under-representation of storage of water) quite useful. So I would advise to move the technical method to the supplement but keep the message in the main body of the manuscript.

OTHER COMMENTS:

1) It must be mentioned in the abstract that the HCLIM-ALARO simulation is driven by ERA-Interim
2) Abstract, L 13: remove "precisely"
3) Abstract, L 14, 18: replace "lack" by "absence" According to the text in Line 99 "the model does not take into account wintertime road maintenance operations …". From that line I conclude that there is no maintenance at all in the model. "Lack" may imply there is still some maintenance left. Please, adjust everywhere in the text, if needed.
4) Abstract, L 17: remove "simulated", it is already implied by "warm bias".

5) Introduction, L 24: "climate and weather information" →"weather and climate information".
6) Introduction, L34: "Finish temperatures …" →" Finish temperature records …"
7) Introduction, L42:  replace "reliable" by "plausible" or "credible". It is not a prediction.
8) Introduction, L65: "13 year long simulations" → "13-year long simulations".

9) Page 3, L85-88: mention the source of the sea-surface boundary conditions (SST and sea-ice extent (probably also ERA-Interim)
10) Page 3, L92: "transfer in the ground …" →"transfer *into* the ground …"
11) Page 4, L95: ".. the elevation is taken into account …" The elevation of what or with respect to what?
12) Page 4, L107: "… we did not include any forecast periods". Suggest to add the phrase "implying that no in-situ observations are used to initialize and force RoadSurf."
13) Page 4, L119: Mention the version of the E-OBS dataset.
14) Page 4, L120: In addition to daily mean temperature, E-OBS also contains daily minimum/maximum temperature. Why not using these parameters for evaluation?
15) Page 4, L 122: remove "some"
16) Page 5, L 127: remove "some"
17) Page 6, L 180: the phrase "… such as from the possible biases in the input parameters ERA-Interim …" is confusing. Do you mean that land-surface information from ERA-Interim is used in forcing HCLIM-ALARO, or does this statement refer to the lateral/sea-surface boundary conditions specified from ERA-Interim?

18) Page 7, L197: "Similarly than in …" →"Similar to …"
19) Page 8, L227: "… during different months" →"… for different months"
20) Page 9, L268: "earlier" → "before"
21) Page 10, L296: "further" → "hence"
22) Page 10, L300-311: "the stations" → "stations" (about 11x)
23) Page 10, L307: "It could be expected …" →"It might be expected …"
24) Page 10, L316: "between the different stations" → "between stations"
25) Page 10, L317: "hypothesized" → "speculated"
26) Page 11, L343: "class occurred within a month" →"class *occurring* within a month"
27) Page 11, section 3.2.4 and Fig. 9 Perhaps you could briefly repeat that the road surface classes in the observations and the model do not entirely match.
28) Page 12, L356-357: rephrase last part of sentence as "i.e., the tendency *of the model* to underestimate frost and to overestimate ice with the same magnitude."
29) Page 12, L360-361: "where much less maintenance" → ""where maintenance" and "is performed compared to …" →" is performed far less frequently compared to …"
30) Page, 12, L 362: "In real life," → "In reality"
31) Page, 12, L 365-367: That is precisely the problem, because the bias in forcing temperature has not been adjusted the distinction between those two error sources cannot be made
32) Page, 12, L 375: No threshold used? Just, plainly 0 when the mean was 0?
33) Page, 12, L 379: "…storages might be slightly displaced or mistimed". That is typical for NWP verification, but should not be relevant in an evaluation study.
34) Page, 14, L417-419: Second part of this sentence, "however …" is unclear. Please rephrase.
35) Conclusions, L 420-423. Like in the abstract it should be stated that HCLIM-ALARO is driven by ERA-Interim re-analyses.
36) Conclusions, L 422: "the skill of HCLIM- …" →"the skill of *the* HCLIM- …"
37) Conclusions, L 427: "undercath" →"undercatch"
38) Conclusions, L 427-428: "the modeled domain" →"the model domain"
39) Conclusions, L 432-433: Remove "However,". Moreover, the absence of data-assimilation is most probably at least as relevant as the difference in horizontal resolution for explaining the poorer performance.

40) Conclusions, L 439: "This is of a great importance" →"This is of great importance"
41) Conclusions, L 439: "… are the most slippery…" →"… are most prone to slippery conditions …"
42) Conclusions, L442: "… than what the observations showed" → " than is indicated by observations"
43) Conclusions, L447: "the 13 year long … period" → "the 13-year long … period".

44) Figure caption 1: Does the displayed domain include or exclude the boundary relaxation zone? How wide is the zone in terms of grid points? The color "yellow" for Northern Finland is very hard to distinguish.

---

## Referee Comment (RC2) · Anonymous Referee #2 · 3 Apr 2019

This paper evaluates the RoadSurf model forced with output from a regional climate model (HARMONIE-Climate). The RoadSurf is used operationally to simulate road conditions for the benefit of the public. Here, the authors extend RoadSurf by forcing it with output from a regional climate model. This successful endeavor then paves the way to make assessments of future road conditions under climate change by forcing RoadSurf with output from a projection-period regional climate simulation.

The paper is easy to read and understand. I am not an expert in road modeling, so it is difficult to criticize anything about the RoadSurf model. I certainly couldn't identify any glaring deficiencies. Much of the paper is devoted to assessing the skill of the

regional climate model. There are biases and problems, as one would expect, but even with these biases, the RoadSurf model is able to reasonably replicate what is observed at the observed road sites. Clearly, it would be even more powerful if the simulation forced with regional climate model output could be compared to results with bias-corrected forcing or local forcing, but that may not really be feasible. So, in the context of the purpose of the paper, which is to assess whether or not RoadSurf forced with a regional climate model has the potential to provide useful information on Road conditions now and in the future, I would say that the authors have demonstrated this to be the case.

So, overall, I find this paper suitable for publication in close to it's current form. Will be interesting to see what happens when they run with climate change scenarios.

---

## Author Comment (AC1) · 7 Jun 2019

Dear Executive Editor,

Thank you for your comments. We have now provided the version numbers of RoadSurf and HARMONIE-Climate in the title and in the manuscript. We have also clarified the availability of the RoadSurf model code.
* * *

---

## Author Comment (AC2) · 7 Jun 2019

Dear Reviewer,

Please find as a supplement our point-by-point response to the Referee comments as well as the modified manuscript showing the changes that were conducted.

Please also note the supplement to this comment:
https://www.geosci-model-dev-discuss.net/gmd-2018-330/gmd-2018-330-AC2-supplement.zip

---

## Author Response (AR1)

**Author response for the reviewers regarding the manuscript *"The road weather model RoadSurf driven by the HARMONIE-Climate regional climate model: evaluation over Finland"***

We thank both reviewers for their comments and suggestions (in blue). Please find our detailed point-by-point responses below (in black).

We have made changes in the manuscript, and the changes are visualized at the end of this document. The pages and line numbers as well as the reference numbers for figures used in this response correspond to the ones used in the updated version.

Anonymous Referee #1

GENERAL COMMENTS:

This article presents an evaluation study across Finland of a 13-year long record of road surface temperatures and road ice, snow, and water storage parameters obtained with a road surface weather model driven by output from a regional climate model operated at 12.5 km resolution. The RCM is in turn forced at its lateral boundaries by atmospheric information from the ERA-Interim re-analysis. The emphasis of the analysis is on the performance of the road surface model compared with observations obtained at 25 road weather stations of which 11 were also equipped with optical sensors to establish the prevailing condition of the road surface.

Overall, the paper is coherently written, but in my opinion the scope is too much from the perspective of an NWP. The entire analysis assumes as if the modelling chain can be one-to-one compared with observations and a statistical machinery is applied resulting in skill scores which one usually sees in assessing the forecast performance of a prediction model. This approach does not match with the purpose of the study to evaluate a model system when operated in climate mode but still using observations (ERA-Interim) to constrain the large-scale model circulation to the observed synoptic-scale structure.

We thank the reviewer for the careful and thorough evaluation of our manuscript. We also apologize that we did not emphasize enough the reasoning behind our study that is to assess if RoadSurf can adequately capture the road weather conditions occurred in the current climate when forced with a reanalysis-driven regional climate model (RCM) HARMONIE-Climate (HCLIM). This is the first time such a modeling chain is evaluated and, therefore, we believe that forcing HCLIM with the ERA-Interim reanalysis product suits well this purpose instead of forcing HCLIM directly with global circulation models (GCMs). When forcing HCLIM by a reanalysis product, the large-scale model circulation is constrained by the observed synoptic-scale, as also mentioned by the reviewer. For example, Kotlarski et al. (2014) state that it is a standard procedure to carry out evaluation experiments using the (close to) perfect boundary settings in RCMs, which means using reanalysis product, such as ERA-Interim, to force the regional climate model in the lateral boundaries as it is done in our study. We have therefore clarified the goals of our study in the introduction (see P2–3 L58–64).

Regarding the NWP perspective, it is true that we have used daily data in the analysis of RoadSurf. On the other hand, the regional climate model HCLIM has been evaluated using standard metrics, mean seasonal biases for temperature and mean seasonal relative biases for precipitation using monthly values over 13 years. The daily time scale was chosen for the RoadSurf evaluation because the daily (and even hourly) temporal scales are the most relevant when studying road weather as also mentioned in the manuscript.

However, it is good to note that we have calculated the metrics using daily data obtained for 13 years separately for each month and taken multi-year monthly means of the daily values. This means that for example the mean biases for road surface temperature and mean daily fractions of road surface classes will be the same regardless the time scale in question (e.g. for the biases, the means of the monthly means of daily mean biases and the means of the monthly mean biases are the same; please see Equation 1 below). To better account for the fact that RoadSurf was forced by an RCM, we have performed a part of the analysis (sections 3.2.1, 3.2.3 and 3.2.4) at a (multi-year) monthly scale. The main conclusions stay the same due to the nature of calculating means as explained by the following equation (example for one month):

$$\frac{\sum_{d=1}^{30}\left(\frac{\sum_{h=1}^{24} M_h - O_h}{24}\right)_d}{30} = \frac{\sum_{d=1}^{30}\left(\frac{\sum_{h=1}^{24} M_h}{24}\right)_d}{30} - \frac{\sum_{d=1}^{30}\left(\frac{\sum_{h=1}^{24} O_h}{24}\right)_d}{30} \qquad (1)$$

where $d$ refers to day, $h$ to hour, $M_h$ to $h$th model value, and $O_h$ to $h$th observed value.

The next step, also mentioned by the authors, will be to run the RCM-RoadSurf modelling system driven by GCM output resulting from transient multi-annual simulation under prescribed emission scenarios. The biases then found will presumably be much larger than seen in this evaluation study, and any performance rating as if it were a prediction model will be deemed meaningless. The primary reason for that is huge biases in circulation and regime statistics in the GCM drivers compared to ERA-Interim. So, the authors can better focus on the role of circulation and regime drivers on the performance of their modelling chain, than focus on skill scores like RMSE and Pearson's correlation coefficients. Eventually, they want to draw credible conclusions how climate change information at the large scale will propagate through their RCM to the RoadSurf model.
We agree with the referee that the performance of RoadSurf that is forced by ERA-Interim-driven HCLIM does not mean that we will obtain exactly the same results when HCLIM is forced by GCMs. However, we think the ERA-Interim evaluation is crucial before continuing to use this method further. If RoadSurf would not perform adequately when the input data is coming from an RCM forced by a reanalysis product (i.e. with perfect boundary settings), we see that it would not be appropriate to analyze the effects of climate change on road weather with this method. In our opinion, the evaluation of RoadSurf that is forced by a GCM-driven HCLIM would be a subject of its own study. However, this aspect will be looked at when RoadSurf's inputs are retrieved from a GCM-driven HCLIM. Please see also our comments above (the first paragraph of our response).
As road surface temperature is the main output parameter in RoadSurf, we have, however, added an analysis of the relationships between the road surface temperature biases and the biases in the input parameters of RoadSurf that are retrieved from HCLIM (please see section 3.2.1 starting from P12 L352).

In addition I would argue that the way the experiment has been set up makes it very difficult to conclude how the shortcomings in performance can be attributed to the model components that are used. Several times the authors mention that an issue might be related either to the warm and/or wet bias in their RCM or to features in the RoadSurf model. In that respect, I am wondering why the authors have not carried out a bias adjustment to HCLIM-ALARO temperature and precipitation which serve as forcings to the RoadSurf model. Such an additional experiment would have a twofold benefit: a) to disentangle the bias in HCLIM from issues in RoadSurf, and b) to obtain a measure to

what extent the biases in HCLIM affect the performance of RoadSurf. The latter would be very helpful in the analysis and interpretation of future GCM-driven experiments.
We agree with the referee that bias-correction could be helpful in distinguishing the biases caused by the input retrieved from HCLIM and by RoadSurf itself. However, we would like to remind that the purpose of this study was to evaluate the whole modeling chain (i.e., the model biases in RoadSurf when forced by reanalysis-driven HCLIM) and to show that RoadSurf can reliably capture occurred road weather conditions in Finland when driven by an RCM.

In addition, we believe that carrying out bias-correction is therefore not in the scope of our study and would require a considerable amount of work, which might not be feasible in the context of this paper. We also see that bias-correction might not be very beneficial as the distributions of e.g. modeled daily minimum, mean, and maximum temperatures as well as precipitation are already relatively close to observations (See Figs. S2 and S3 in the updated manuscript). In addition, it might not always be very straightforward to use bias-correction, such as quantile mapping (that was suggested by the reviewer), for the future simulations as the error correction values might not be stationary (see e.g. Switanek et al., 2017). Moreover, e.g. Maraun et al. (2017) state that if regional feedbacks are not properly taken into account, bias-correction methods, such as quantile mapping, might lead to implausible regional climate change signals. Another problem can be the physical inconsistencies in the bias-corrected data (Schoetter et al., 2012).

But as said earlier, we have now added an analysis of the relationships between the road surface temperature biases and the biases in the input parameters at the road weather stations. Based on this analysis, the road surface temperature bias seems to be mainly explained by the variability of the air temperature bias (section 3.2.1 starting from P12 L352) as speculated in the first version of the manuscript.

MAJOR COMMENTS:
1) I would strongly suggest to focus on the Finland area from the beginning. The discussion of the HCLIM-ALARO model performance for the whole of the Fenno-Scandian domain is distracting. There are always huge issues in the mountainous areas in Norway, for any RCM, and also in E-OBS, but they are not relevant for this study. Focus on Finland in Figs 3 and 5.
Thank you for the suggestion. In the updated manuscript, we show results only over Finland (please see Figs. 3 and 6). Also, the discussion is now adjusted to cover only Finland for section "3.1 Evaluation of HCLIM38-ALARO".

2) Do not only examine the bias in the monthly mean temperature, but also at a number of percentiles (e.g. P5,25,75,95). The diurnal amplitude in model temperature compared to observations is relevant here as well.
The temperature percentiles (P5, P22, P75 & P95) have been added in the Supplement (Fig. S5) and are discussed in the new section 3.1.2. Moreover, we have included figures of the biases in daily minimum and maximum temperatures to account for the diurnal cycle (Fig. 5 in the new section 3.1.2).

It is still good to note that the purpose of our paper is not to evaluate very thoroughly the performance of HCLIM, but rather focus on the performance of RoadSurf.

3) Similarly for precipitation. In addition to mean precipitation look at wet-day frequency (threshold 0.3 or 1.0 mm/day), and perhaps some exceedance percentiles. It provides much more insight than an RMSE score.
Initially, we did not use RMSE scores for precipitation, but showed mean relative seasonal precipitation biases (Figs. 6 and 7). We have included a new figure on the wet-day frequency (Fig. S6) and added a discussion regarding this figure in section 3.1.3.

As said above, it is good to keep in mind that the purpose of our paper is not to evaluate very thoroughly the performance of HCLIM, but rather focus on the performance of RoadSurf.

4) Can there be said anything about the accuracy of the RCM inputs other than near-surface temperature and precipitation that are used to drive the RoadSurf model.
We have now added a brief evaluation of other input parameters (relative humidity, wind speed, as well as shortwave and longwave radiation) by comparing HCLIM model results and ERA5 reanalysis product (please see section 3.1.4). In addition, we have briefly evaluated the modeled total cloud fraction.

5) As mentioned in the general comments it would be useful to apply a bias-adjustment on daily mean temperature and precipitation, also frequency of occurrence, to bring the HCLIM-ALARO temperature (e.g. quantile-quantile) and precipitation forcing in the same "statistical" ballpark as the observations.
Please see our response above (the last paragraph of general comments).

6) As the RCM is operated at 12.5km resolution there should be reference to the efforts within EuroCordex in conducting evaluation (ERA-Interim driven) and transient (GCM driven) experiments at 12.5 km resolution across Europe with a variety of RCMs. For the evaluation study you best cite Kotlarski et al. (2014; doi:10.5194/gmd-7-1297-2014).
Thank you for pointing out this relevant reference. The EURO-CORDEX initiative has been mentioned in the introduction as well as in the discussion of the results obtained with HCLIM.

The following phrases were added:

"*Although high-resolution climate projections for Europe are currently available through the international climate downscaling initiative EURO-CORDEX that provides RCM data at 50 km (EUR-44) and 12.5 km (EUR-11) resolution (Jacob et al., 2014), the EURO-CORDEX dataset does not publicly include reanalysis-driven RCM simulations at very high temporal resolutions, such as 1-hourly.*" (P3 L64–67)

"*On the other hand, the HCLIM38-ALARO results for mean seasonal $T_{air}$ were in agreement with EURO-CORDEX RCMs that were run at 12.5 km grid resolution. For instance, Kotlarski et al. (2014) showed that some of the ERA-Interim-driven EURO-CORDEX RCMs had a warm (cold) bias especially over the northern parts of Finland during the winter (summer).*" (P8 L230–233)

"*The results obtained for HCLIM38-ALARO showed similar magnitude and spatial patterns of the precipitation biases compared to other EURO-CORDEX RCMs that are mainly overestimating seasonal precipitation over Finland during the winter and summer as shown by Kotlarski et al. (2014).*" (P9 L282–285)

"*Overall, the HCLIM38-ALARO results were found to be in line with other EURO-CORDEX RCMs.*" (P19 L587–588)

7) Section 3.2.1 ("Road surface temperature"), after line 240 bothers me most. Why are all discrepancies blamed on the bias in temperature forcing, and not on potential issues with downwelling radiation, in particular biases in downwelling long wave radiation due to biases in cloud amount or cloud base.

We have now evaluated the biases in the downwelling radiation (both shortwave and longwave). The variability in the biases of downwelling longwave radiation seems to play a small role in explaining the variability of road surface temperature biases. However, the biases in the air temperature have a much larger impact (please see section 3.2.1 starting from P12 L352) as speculated in the first version of the manuscript.

The following phrases were added:

"*The analysis shown in Fig. 10 revealed that the variability of the monthly biases in $T_{air}$ explained on average 57 % (range 19–84 % in October–April) of the variability of the monthly biases in $T_{road}$ while the $LW_d$ biases explained on average 16 % (range 2–34 % in October–March). Furthermore, the variability in $SW_d$ biases was found to explain a small amount (4 %) of the variability in $T_{road}$ biases during April.*" (P12 L355–358)

8) Page 9, L260-266. The authors argue that the better skill obtained with the forcing from the NWP compared to this study can be ascribed to the higher resolution at which the NWP is operated. I tend to disagree on that, in my opinion the use of data-assimilation when operating in NWP-mode will keep the model atmospheric state across the Finland region much closer to the observed state.

Thank you for pointing this out. However, we have compared the results only to the ones that are obtained without any data assimilation. This has been clarified in the text as follows:

"*For example, Karsisto et al. (2016) found that the biases in the simulated $T_{road}$ varied between –1 and 1 ºC (mostly ±2 ºC in our study) at 20 stations in Finland during October and December 2013 when RoadSurf was driven by a high-resolution NWP version of HARMONIE (cy36h1.4) with a grid resolution of 2.5 km without any data assimilation. However, it is good to note that the results obtained in our study and by Karsisto et al. (2016) are not directly comparable since in their study RoadSurf was initialized using road weather station measurements for 48 hours and only the first forecasted hour was analyzed.*" (P13  L392–398).

9) The statistical methods used in sections 3.2.2. are not suitable for evaluation purposes, they belong to the realm of NWP verification. I advise to take this section out or move it to the supplement.

This old section 3.2.2 has been removed, and some parts of the discussion on the model performance from this section have been moved to section 3.2.1 (starting from P12 L382).

10) The same applies to section 3.2.5 although I find the message (i.e. over-representing of storage of ice, under-representation of storage of water) quite useful. So I would advise to move the technical method to the supplement but keep the message in the main body of the manuscript.

The technical method has been moved to the supplement, and only the results from POD-FAR analysis are kept (see section 3.2.4). In addition, we have added a figure and discussion on the multi-year sums of the occurrence of the storages for both model and observations thus avoiding a day-to-day comparison (Fig. S7; please see section 3.2.4 starting from P17 L528).

OTHER COMMENTS:

1) It must be mentioned in the abstract that the HCLIM-ALARO simulation is driven by ERA-Interim

This is now mentioned in the abstract as follows:

"*RoadSurf was driven by meteorological input data from the cycle 38 of the high-resolution regional climate model (RCM) HARMONIE-Climate (HCLIM38) with ALARO physics (HCLIM38-ALARO) and ERA-Interim forcing in the lateral boundaries.*" (P1 L10–11).

2) Abstract, L 13: remove "precisely"
Removed (P1 L15).

3) Abstract, L 14, 18: replace "lack" by "absence" According to the text in Line 99 "the model does not take into account wintertime road maintenance operations ...". From that line I conclude that there is no maintenance at all in the model. "Lack" may imply there is still some maintenance left. Please, adjust everywhere in the text, if needed.
Corrected (P1 L16, P1 L22, P4 L119, P16 L501, P18 L553, P18 L557, & P19 L608).

4) Abstract, L 17: remove "simulated", it is already implied by "warm bias".
Removed (P1 L19).

5) Introduction, L 24: "climate and weather information" → "weather and climate information".
Corrected (P1 L28).

6) Introduction, L34: "Finish temperatures ..." → " Finish temperature records ..."
Corrected as "Finnish temperature records" (P2 L38).

7) Introduction, L42: replace "reliable" by "plausible" or "credible". It is not a prediction.
Changed as "credible" (P2 L46)

8) Introduction, L65: "13 year long simulations" → "13-year long simulations".
Corrected (P3 L78).

9) Page 3, L85-88: mention the source of the sea-surface boundary conditions (SST and sea-ice extent (probably also ERA-Interim)
Yes, these parameters (SST and sea-ice concentration) are taken from ERA-Interim. This information has been added in the text as follows:

"*The sea-surface (sea-surface temperature and sea-ice concentration) and lateral boundary conditions of HCLIM38-ALARO were taken from ERA-Interim reanalysis (Dee et al., 2011) every 6 hours.*" (P4 L102–104).

10) Page 3, L92: "transfer in the ground ..." → "transfer into the ground ..."
Corrected (P4 L110).

11) Page 4, L95: ".. the elevation is taken into account ..." The elevation of what or with respect to what?
By elevation, we mean topography in general as RoadSurf otherwise assumes a flat surface. Modified as follows:

"*However, topography in general is taken into account implicitly through the input data.*" (P4 L114).

12) Page 4, L107: "... we did not include any forecast periods". Suggest to add the phrase "implying that no in-situ observations are used to initialize and force RoadSurf."
This has been added in the text (P4 L125–126).

13) Page 4, L119: Mention the version of the E-OBS dataset.
In the first version of the manuscript, we used the E-OBS version 17.0. In the updated manuscript, we have utilized the minimum and maximum temperatures provided by E-OBS version 19.0e (as suggested in the reviewer comment 14). Thus, we redid the analysis of mean temperatures and precipitation using the new ensemble version 19.0e. This information is now added in the text (P5 L140). In addition, we updated all the figures where E-OBS data was used (Figs. 3–4 & 6–7).

14) Page 4, L120: In addition to daily mean temperature, E-OBS also contains daily minimum/maximum temperature. Why not using these parameters for evaluation?
The minimum and maximum temperatures provided by the E-OBS version 19.0e have now been employed in the evaluation of HCLIM38-ALARO (please see section 3.1.2).

15) Page 4, L 122: remove "some"
Removed (P5 L146).

16) Page 5, L 127: remove "some"
Removed (P5 L151).

17) Page 6, L 180: the phrase "... such as from the possible biases in the input parameters ERA-Interim ..." is confusing. Do you mean that land-surface information from ERA-Interim is used in forcing HCLIM-ALARO, or does this statement refer to the lateral/sea-surface boundary conditions specified from ERA-Interim?
ERA-Interim's SST and sea-ice concentration are used to force HCLIM. If these input parameters from ERA-Interim are biased, it might affect the performance of HCLIM. This has been clarified in the text. This matter would need further evaluation, which is unfortunately not in the scope of this study.

The following phrases were added:
"*A prognostic lake model was included in the model version used in this study, and thus the warm bias might have stemmed from other reasons, such as from SURFEX's own features or the possible biases in ERA-Interim's sea-surface temperatures or sea-ice concentrations that are used to force the sea-surface in HCLIM.*" (P8 L227–230)

18) Page 7, L197: "Similarly than in ..." → "Similar to ..."
Corrected (P9 L269).

19) Page 8, L227: "... during different months" → "... for different months"
Corrected (P11 L335).

20) Page 9, L268: "earlier" → "before"
This part of the old section 3.2.2 is removed, and therefore this correction was not made.

21) Page 10, L296: "further"→"hence"
Corrected (P13 L386).

22) Page 10, L300-311: "the stations"→"stations" (about 11x)
This part of the old section 3.2.2 is removed, and therefore this correction was not made.

23) Page 10, L307: "It could be expected ..." → "It might be expected ..."
This part of the old section 3.2.2 is removed, and therefore this correction was not made.

24) Page 10, L316: "between the different stations" → "between stations"
This part of the old section 3.2.2 is removed, and therefore this correction was not made.

25) Page 10, L317: "hypothesized" → "speculated"
This part of the old section 3.2.2 is removed, and therefore this correction was not made.

26) Page 11, L343: "class occurred within a month" → "class occurring within a month"
Corrected (P15 L483).

27) Page 11, section 3.2.4 and Fig. 9 Perhaps you could briefly repeat that the road surface classes in the observations and the model do not entirely match.
This information has been repeated as follows:

"*It is good to remember that the observed and modeled road surface classes might not fully match as they are defined differently.*" (P16 L485–486).

"*The definitions of road surface classes differ slightly for the observations and model (e.g. the partly icy class is included only in the model).*" (Fig. 12; P35 L1085–1086)

28) Page 12, L356-357: rephrase last part of sentence as "i.e., the tendency of the model to underestimate frost and to overestimate ice with the same magnitude."
Rephrased as suggested (P16 L499–500).

29) Page 12, L360-361: "where much less maintenance" → "where maintenance" and "is performed compared to ..." → " is performed far less frequently compared to …"
Corrected (P16 L502–504).

30) Page, 12, L 362: "In real life," → "In reality"
Corrected (P16 L505).

31) Page, 12, L 365-367: That is precisely the problem, because the bias in forcing temperature has not been adjusted the distinction between those two error sources cannot be made
We are discussing the possible reasons for the bias, so we would not call this a problem but rather a speculation. Most likely, both factors are contributing to the errors. No changes made.

32) Page, 12, L 375: No threshold used? Just, plainly 0 when the mean was 0?
Correct. This is because the daily mean values for storages were very small and very often the storages are plainly zero. We tested the same method using daily median and daily maximum values, but this did not considerably affect the results. However, we decided to use daily maximum values in the updated manuscript to avoid any confusion (see section 3.2.4).

33) Page, 12, L 379: "...storages might be slightly displaced or mistimed". That is typical for NWP verification, but should not be relevant in an evaluation study.
This part has been modified as follows:

"*However, this method might penalize the model more than it should because the modeled storages were compared with observations using day-to-day values. For this reason, we additionally calculated the multi-year sums of all the modeled and observed daily cases with daily maximum more than zero or zero.*" (P17 L526–529).

34) Page, 14, L417-419: Second part of this sentence, "however ..." is unclear. Please rephrase.
Rephrased as:

"*However, underestimated frequency of snow cannot be explained by the snowpacks that are depleting too fast in the model. This is because the majority of the stations with an optical sensor utilized in this study are located in the southern parts of Finland where the modeled snowpacks might actually stay longer compared to the measurements as discussed before.*" (P18 L569–571).

35) Conclusions, L 420-423. Like in the abstract it should be stated that HCLIM-ALARO is driven by ERA-Interim re-analyses.
Thank you for pointing this out – this statement is included (P18 L580–581).

36) Conclusions, L 422: "the skill of HCLIM- ..." → "the skill of the HCLIM- ..."
Corrected (P18 L579).

37) Conclusions, L 427: "undercath" → "undercatch"
Corrected (P19 L587).

38) Conclusions, L 427-428: "the modeled domain" → "the model domain"
Changed as "Finland" (P19 L587).

39) Conclusions, L 432-433: Remove "However,". Moreover, the absence of data-assimilation is most probably at least as relevant as the difference in horizontal resolution for explaining the poorer performance.
This sentence is modified as:

"*The coarser grid resolution of the HCLIM38-ALARO compared to the NWP model input used in the earlier studies might be the main reason for this outcome as no data assimilation was used for HCLIM38-ALARO or the NWP model.*" (P19 L595–597).

40) Conclusions, L 439: "This is of a great importance" → "This is of great importance"
Corrected (P19 L604).

41) Conclusions, L 439: "... are the most slippery..." → "... are most prone to slippery conditions ..."
Corrected (P19 L604).

42) Conclusions, L442: "... than what the observations showed" → " than is indicated by observations"
Corrected as "*than indicated by observations*" (P19 L607–608).

43) Conclusions, L447: "the 13 year long ... period" → "the 13-year long ... period".
Corrected (P19 L612).

44) Figure caption 1: Does the displayed domain include or exclude the boundary relaxation zone? How wide is the zone in terms of grid points? The color "yellow" for Northern Finland is very hard to distinguish.
The extension zone of 11 grid points has been removed from the figure. However, the figure includes an 8-point wide relaxation zone. This information has been added to the figure caption (see Fig. 1). The colors are changed for Fig. 1 and 2, so that yellow color is not used.

The following phrases were added:
"*Figure 1 depicts the HCLIM38-ALARO simulated domain along with the model's 8-point wide relaxation zone as well as the regions of Finland that are analyzed in more detail in this study.*" (P4 L101–102)

"*The transparent areas depict the model's 8-point wide relaxation zone.*" (Fig. 1; P26 L826)

Anonymous Referee #2

This paper evaluates the RoadSurf model forced with output from a regional climate model (HARMONIE-Climate). The RoadSurf is used operationally to simulate road conditions for the benefit of the public. Here, the authors extend RoadSurf by forcing it with output from a regional climate model. This successful endeavor then paves the way to make assessments of future road conditions under climate change by forcing RoadSurf with output from a projection-period regional climate simulation.

The paper is easy to read and understand. I am not an expert in road modeling, so it is difficult to criticize anything about the RoadSurf model. I certainly couldn't identify any glaring deficiencies. Much of the paper is devoted to assessing the skill of the regional climate model. There are biases and problems, as one would expect, but even with these biases, the RoadSurf model is able to reasonably replicate what is observed at the observed road sites. Clearly, it would be even more powerful if the simulation forced with regional climate model output could be compared to results with bias-corrected forcing or local forcing, but that may not really be feasible. So, in the context of the purpose of the paper, which is to assess whether or not RoadSurf forced with a regional climate model has the potential to provide useful information on Road conditions now and in the future, I would say that the authors have demonstrated this to be the case.

So, overall, I find this paper suitable for publication in close to it's current form. Will be interesting to see what happens when they run with climate change scenarios.

We thank the referee for the positive feedback on our manuscript. Evaluation of RoadSurf using local forcing would be interesting, but this is not feasible as the road weather stations in Finland do not observe solar radiation and precipitation measurements are considered unreliable (Kangas et al., 2015).

However, we have now added an analysis of the relationships between the road surface temperature biases and the biases in the input parameters at the road weather stations. Based on this analysis, the variability in the road surface temperature biases seems to be mainly explained by the variability in the air temperature biases (please see section 3.2.1 starting from P12 L352) as speculated in the first version of the manuscript.

[revised manuscript text omitted]
 a̶u̶t̶u̶m̶n̶the winter (̶S̶e̶p̶t̶e̶m̶b̶e̶r̶–̶O̶c̶t̶o̶b̶e̶r̶)̶ with a domain-averaged bias of 1̶2̶.̶7̶16.1 % and highest in the spring (̶M̶a̶r̶c̶h̶–̶M̶a̶y̶)̶ with a domain-averaged bias of 3̶1̶.̶9̶42.2 %. The h̶i̶g̶h̶e̶s̶t̶ largest biases in simulated precipitation occurred in t̶h̶e̶ ̶N̶o̶r̶w̶e̶g̶i̶a̶n̶ ̶m̶o̶u̶n̶t̶a̶i̶n̶s̶the north of Finland, especially over Lapland, where the biases were also statistically significant for every season. ̶a̶s̶ ̶w̶e̶l̶l̶ ̶a̶s̶ ̶o̶v̶e̶r̶ ̶t̶h̶e̶ ̶m̶o̶u̶n̶t̶a̶i̶n̶o̶u̶s̶ ̶a̶r̶e̶a̶s̶,̶ ̶w̶h̶i̶c̶h̶ ̶m̶a̶y̶ ̶p̶e̶n̶a̶l̶i̶z̶e̶ ̶t̶h̶e̶ ̶m̶o̶d̶e̶l̶ ̶i̶n̶ ̶t̶h̶e̶ ̶a̶r̶e̶a̶s̶ ̶w̶i̶t̶h̶ ̶t̶h̶e̶ ̶m̶o̶s̶t̶ ̶c̶o̶m̶p̶l̶e̶x̶ ̶t̶o̶p̶o̶g̶r̶a̶p̶h̶y̶.̶ ̶W̶e̶ ̶s̶t̶r̶e̶s̶s̶ ̶t̶h̶a̶t̶ ̶E̶-̶O̶B̶S̶ ̶m̶i̶g̶h̶t̶ ̶s̶u̶f̶f̶e̶r̶ ̶f̶r̶o̶m̶ ̶u̶n̶d̶e̶r̶c̶a̶t̶c̶h̶ ̶e̶r̶r̶o̶r̶s̶ ̶d̶u̶r̶i̶n̶g̶ ̶t̶h̶e̶ ̶w̶i̶n̶t̶e̶r̶ ̶a̶n̶d̶ ̶s̶p̶r̶i̶n̶g̶.̶ The biases were statistically significant over the whole model domain during the spring and summer season. We stress that E-OBS might suffer from undercatch errors during the winter and spring. The larger biases in the northern parts of Finland might again originate from the sparser observation network in the northernmost domain.D̶u̶r̶i̶n̶g̶ ̶t̶h̶e̶ ̶w̶i̶n̶t̶e̶r̶ ̶a̶n̶d̶ ̶a̶u̶t̶u̶m̶n̶ ̶s̶e̶a̶s̶o̶n̶s̶,̶ ̶t̶h̶e̶ ̶b̶i̶a̶s̶e̶s̶ ̶w̶e̶r̶e̶ ̶s̶i̶g̶n̶i̶f̶i̶c̶a̶n̶t̶ ̶m̶a̶i̶n̶l̶y̶ ̶i̶n̶ ̶t̶h̶e̶ ̶n̶o̶r̶t̶h̶e̶r̶n̶ ̶p̶a̶r̶t̶s̶ ̶o̶f̶ ̶t̶h̶e̶ ̶m̶o̶d̶e̶l̶ ̶d̶o̶m̶a̶i̶n̶ ̶(̶e̶.̶g̶.̶ ̶t̶h̶e̶ ̶n̶o̶r̶t̶h̶e̶r̶n̶m̶o̶s̶t̶ ̶F̶i̶n̶l̶a̶n̶d̶)̶ ̶a̶n̶d̶ ̶i̶n̶ ̶L̶a̶t̶v̶i̶a̶ ̶i̶n̶ ̶a̶d̶d̶i̶t̶i̶o̶n̶ ̶t̶o̶ ̶N̶o̶r̶w̶a̶y̶.̶ ̶A̶g̶a̶i̶n̶,̶ ̶s̶o̶m̶e̶ ̶p̶a̶r̶t̶ ̶o̶f̶ ̶t̶h̶e̶ ̶b̶i̶a̶s̶e̶s̶ ̶m̶i̶g̶h̶t̶ ̶h̶a̶v̶e̶ ̶b̶e̶e̶n̶ ̶c̶a̶u̶s̶e̶d̶ ̶b̶y̶ ̶t̶h̶e̶ ̶l̶a̶c̶k̶ ̶o̶f̶ ̶a̶ ̶d̶e̶n̶s̶e̶ ̶o̶b̶s̶e̶r̶v̶a̶t̶i̶o̶n̶ ̶n̶e̶t̶w̶o̶r̶k̶ ̶i̶n̶ ̶t̶h̶e̶ ̶n̶o̶r̶t̶h̶e̶r̶n̶m̶o̶s̶t̶ ̶d̶o̶m̶a̶i̶n̶.̶ ̶S̶t̶a̶t̶i̶s̶t̶i̶c̶a̶l̶l̶y̶ ̶s̶i̶g̶n̶i̶f̶i̶c̶a̶n̶t̶ ̶d̶i̶f̶f̶e̶r̶e̶n̶c̶e̶s̶ ̶d̶u̶r̶i̶n̶g̶ ̶t̶h̶e̶ ̶s̶p̶r̶i̶n̶g̶ ̶s̶e̶a̶s̶o̶n̶ ̶o̶c̶c̶u̶r̶r̶e̶d̶ ̶a̶l̶m̶o̶s̶t̶ ̶i̶n̶ ̶t̶h̶e̶ ̶w̶h̶o̶l̶e̶ ̶F̶i̶n̶l̶a̶n̶d̶,̶ ̶t̶h̶e̶ ̶n̶o̶r̶t̶h̶e̶r̶n̶ ̶p̶a̶r̶t̶ ̶o̶f̶ ̶E̶u̶r̶o̶p̶e̶a̶n̶ ̶R̶u̶s̶s̶i̶a̶,̶ ̶n̶o̶r̶t̶h̶e̶r̶n̶ ̶S̶w̶e̶d̶e̶n̶,̶ ̶p̶a̶r̶t̶l̶y̶ ̶t̶h̶e̶ ̶B̶a̶l̶t̶i̶c̶ ̶c̶o̶u̶n̶t̶r̶i̶e̶s̶,̶ ̶a̶n̶d̶ ̶N̶o̶r̶w̶a̶y̶.̶ 
[revised manuscript text omitted]

1125